# The anti-tubercular drug delamanid as a potential oral treatment for visceral leishmaniasis

Stephen Patterson[1,2†], Susan Wyllie[1†], Suzanne Norval[1], Laste Stojanovski[1,2], Frederick RC Simeons[1,2], Jennifer L Auer[1‡], Maria Osuna-Cabello[1,2], Kevin D Read[1,2], Alan H Fairlamb[1,2*]

[1]Division of Biological Chemistry and Drug Discovery, School of Life Sciences, University of Dundee, Dundee, United Kingdom; [2]Drug Discovery Unit, School of Life Sciences, University of Dundee, Dundee, United Kingdom

**Abstract** There is an urgent requirement for safe, oral and cost-effective drugs for the treatment of visceral leishmaniasis (VL). We report that delamanid (OPC-67683), an approved drug for multi-drug resistant tuberculosis, is a potent inhibitor of *Leishmania donovani* both in vitro and in vivo. Twice-daily oral dosing of delamanid at 30 mg kg$^{-1}$ for 5 days resulted in sterile cures in a mouse model of VL. Treatment with lower doses revealed a U-shaped (hormetic) dose-response curve with greater parasite suppression at 1 mg kg$^{-1}$ than at 3 mg kg$^{-1}$ (5 or 10 day dosing). Dosing delamanid for 10 days confirmed the hormetic dose-response and improved the efficacy at all doses investigated. Mechanistic studies reveal that delamanid is rapidly metabolised by parasites via an enzyme, distinct from the nitroreductase that activates fexinidazole. Delamanid has the potential to be repurposed as a much-needed oral therapy for VL.

*For correspondence: a.h. fairlamb@dundee.ac.uk

†These authors contributed equally to this work

Present address: ‡Department of Chemistry, Kings College London, London, United Kingdom

Competing interests: The authors declare that no competing interests exist.

## Introduction

The repurposing of drugs and clinical candidates offers an attractive alternative to de novo drug discovery (*Fischbach and Walsh, 2009*; *Cragg et al., 2014*; *Peters, 2013*; *Law et al., 2013*; *Novac, 2013*; *Aube, 2012*), particularly in terms of reducing research and development costs for neglected diseases of poverty (*Andrews et al., 2014*). Visceral leishmaniasis (VL), a neglected tropical disease resulting from infection with the protozoan parasites *Leishmania donovani* or *L. infantum* is a case in point, with the two anti-leishmanial front-line therapies miltefosine and amphotericin B both originally developed for other indications (*Stuart et al., 2008*). In addition, the anti-trypanosomal clinical candidate fexinidazole was recently discovered to have potent activity in a murine VL model (*Wyllie et al., 2012*), resulting in a phase II proof of concept clinical trial (NCT01980199) against VL being conducted in Sudan.

There are approximately 50,000 reported cases of VL per year, with the vast majority of infections in South America, East Africa and the Indian subcontinent. However, the number of cases is likely to be vastly underreported, with the actual annual incidence estimated to be between 200,000 and 400,000 (*Alvar et al., 2012*). VL is fatal if untreated and, in the absence of effective vaccines and vector control methods, efficacious chemotherapy is required to combat the disease. Each of the currently available drugs has one or more drawbacks, including the need for hospitalization, prolonged therapy, parenteral administration, high cost, variable efficacy, severe toxic side-effects and resistance (*Croft et al., 2006*). Thus, there is an urgent need for better, safer efficacious drugs that are fit-for-purpose in resource-poor settings.

**eLife digest** Better, safer, oral drugs are desperately needed for the treatment of visceral leishmaniasis, a parasitic infectious disease that causes an estimated 40,000 deaths a year, predominantly in South America, East Africa and the Indian subcontinent. The parasite that causes visceral leishmaniasis is transmitted between individuals by blood-sucking sandflies, and there are currently no vaccines that protect against the disease. In addition, all currently available drug treatments have serious limitations – they are expensive, toxic, have to be applied over a long period of time (mainly by injection) and may become ineffective as the parasites adapt to resist the drug.

A cost-effective way to find a new treatment for a disease is to repurpose existing clinically approved drugs that are used to treat other diseases. Patterson, Wyllie et al. now report that a drug called delamanid, which was recently approved for the treatment of tuberculosis, can cure visceral leishmaniasis in mice. The drug worked when applied orally at doses that might be achievable in human patients, and can also kill parasites obtained from human patients.

Patterson, Wyllie et al. also provide evidence that suggests that delamanid is processed in the parasites by an unknown enzyme. However, this enzyme is not the one that activates a different class of drugs that are used to treat visceral leishmaniasis. Future studies now need to identify the enzyme that is targeted by delamanid, and could investigate combinations of drugs that slow the emergence of resistant parasites and improve delamanid's safety and effectiveness. Clinical trials are required to test how well delamanid treats visceral leishmaniasis in humans.

Given the success of repurposing fexinidazole for use in the treatment of VL (*Wyllie et al., 2012*), there is now a renewed interest in the anti-parasitic potential of nitroaromatic drugs. Recently, we demonstrated that the anti-tubercular clinical candidate (*S*)-PA-824 possesses moderate activity against *L. donovani* parasites both in vitro and in vivo (*Patterson et al., 2013*). Although (*R*)-PA-824, the enantiomer of the candidate showed superior activity, this compound has not entered pre-clinical development, precluding a rapid move to a VL clinical trial. In addition, a recently reported screen of anti-tubercular nitroimidazoles against *L. donovani* identified DNDI-VL-2098 as a suitable compound for further preclinical evaluation (*Mukkavilli et al., 2014*; *Gupta et al., 2015*). The high degree of structural similarity between delamanid (Deltyba, OPC-67683) and both (*R*)-PA-824 and DNDI-VL-2098 (*Figure 1*) prompted us to investigate this nitroimidazole, which has recently received conditional approval in Europe for the treatment of multidrug-resistant tuberculosis (*Committee for Medicinal Products for Human Use, 2013*; *Ryan and Lo, 2014*).

## Results

### In vitro sensitivity of *L. donovani* to (*S*)- and (*R*)-delamanid

The life cycle of *L. donovani* alternates between a flagellated promastigote form residing in the alkaline midgut of the female sandfly vector and an amastigote form that multiplies intracellularly in acidic phagolysosomes of the mammalian host macrophages. Both stages can be cultured axenically; however, intra-macrophage cultures of amastigotes are a more suitable model of mammalian infection for drug discovery. The anti-tubercular drug delamanid and its corresponding *S*-enantiomer were synthesized (Appendix 1 and *Figure 1—figure supplement 1*) and assessed for anti-leishmanial activity. The potency of both compounds was determined in vitro against *L. donovani* (LdBOB) promastigotes and against intracellular amastigotes (LV9) in mouse peritoneal macrophages. The (*S*)-enantiomer of delamanid showed promising anti-leishmanial activity against both developmental stages of the parasite (EC$_{50}$ values of 147 ± 4 and 1332 ± 106 nM against promastigotes and amastigotes, respectively). However, delamanid (the *R*-enantiomer) proved to be an order of magnitude more potent against promastigotes, axenic amastigotes and intracellular amastigotes with EC$_{50}$ values of 15.5, 5.4 and 86.5 nM, respectively (*Table 1*). Both compounds were found to be inactive (EC$_{50}$ >50 µM) in a counter screen against the mammalian cell line HepG2 (*Table 1*).

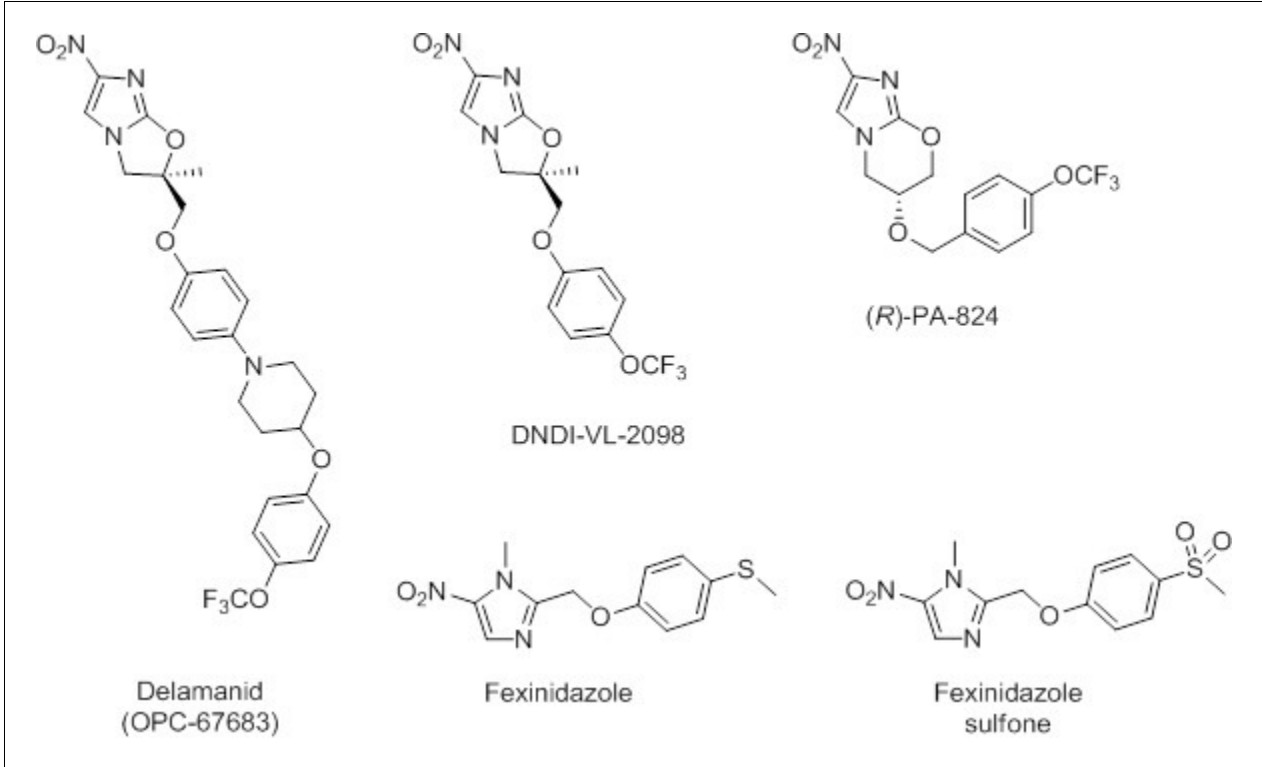

**Figure 1.** Chemical structures of delamanid ((*R*)-OPC-67683) and the known anti-leishmanial nitroimidazoles DNDI-VL-2098, (*R*)-PA-824, fexinidazole and fexinidazole sulfone. Synthetic schemes for the synthesis of delamanid and analogues are described in *Figure 1—figure supplements 1–2*.

The following figure supplements are available for figure 1:

**Figure supplement 1.** Synthetic route towards delamanid (7) and (*S*)-delamanid (13).

**Figure supplement 2.** Synthetic route towards des-nitro-delamanid (18).

Future anti-leishmanial therapies will be required to demonstrate a broad spectrum of activity against different *Leishmania* strains and against drug resistant parasites (*Patterson and Wyllie, 2014*). With this in mind, *L. donovani* and *L. infantum* clinical isolates were assessed for their sensitivity to delamanid (*Table 1*). These included: the Indian WHO reference strain DD8; an Indian antimony resistant isolate BHU1; a recent Sudanese isolate SUKA 001; and the *L. infantum* strain ITMAP263 from Morocco. These clinical isolates were marginally less sensitive to delamanid than our laboratory strain LV9 from Ethiopia, but at the $EC_{90}$ varied by only 3-fold (*L. donovani)* or 8-fold (*L. infantum*) (*Table 1*). Although not investigated further here, promastigotes of *L. major* Friedlin, a cause of cutaneous leishmaniasis, were also highly sensitive to delamanid ($EC_{50}$ 6.3 ± 0.11 nM, slope factor 2.2).

The corresponding des-nitro analogue was also synthesized (Appendix 1 and *Figure 1—figure supplement 2*) and assayed against *L. donovani* promastigotes. Des-nitro-delamanid was found to be inactive ($EC_{50}$ >50 μM), which is consistent with the nitro group being involved in the mechanism of action or having a role in the binding of delamanid to its molecular target(s) in *L. donovani*.

## Physicochemical properties of delamanid

The plasma protein binding of delamanid was measured and found to be high ($F_u$ = 0.0045), in agreement with that reported previously (*Committee for Medicinal Products for Human Use, 2013*). A kinetic solubility assay demonstrated that delamanid possesses sufficient aqueous solubility (>250 μM in 2.5% DMSO) for use in in vitro assays.

**Table 1.** Activity of delamanid against laboratory and clinical isolates of *L. donovani* in vitro. $EC_{90}$ values are calculated from the $EC_{50}$, Hill slopes and the molecular weight of delamanid.

| Species | Developmental stage | $EC_{50}$, nM (Hill slope) | $EC_{90}$, nM | $EC_{90}$, ng ml$^{-1}$ |
|---|---|---|---|---|
| *Leishmania donovani* (LdBOB) | Promastigote | 15.5 ± 0.07 (8.4) | 20.2 | 10.8 |
| *Leishmania donovani* (LdBOB) | Amastigote (axenic) | 5.4 ± 0.05 (5.3) | 8.2 | 4.4 |
| *Leishmania donovani* (LV9) | Amastigote (in macrophage) | 86.5 ± 1.7 (2.3) | 225 | 120 |
| *Leishmania donovani* (DD8) | Amastigote (in macrophage) | 298 ± 13 (2.7) | 672 | 359 |
| *Leishmania donovani* (BHU1) | Amastigote (in macrophage) | 230 ± 10 (4.1) | 393 | 210 |
| *Leishmania donovani* (SUKA001) | Amastigote (in macrophage) | 259 ± 7 (3.6) | 476 | 254 |
| *L. infantum* (ITMAP263) | Amastigote (in macrophage) | 940 ± 0.05 (3.4) | 1790 | 955 |
| Human (HepG2) | N/A | >5000 | - | - |

## Efficacy of delamanid in a murine model of visceral leishmaniasis

The efficacy of delamanid was assessed in a murine model of VL. Groups of infected BALB/c mice (seven days post infection with *ex vivo L. donovani* LV9 amastigotes) were dosed twice-daily, for five consecutive days with an oral formulation of delamanid (1, 3, 10, 30 or 50 mg kg$^{-1}$). On day 14 post-infection, the parasite burdens in the livers of infected mice were determined and compared with those of control animals. The only current oral anti-leishmanial therapy miltefosine (30 mg kg$^{-1}$, once-daily, 5 days) was included as a positive control. Both delamanid and miltefosine were well tolerated at these doses, with no mice displaying any overt signs of toxicity. An initial experiment showed that treatment with delamanid at 50 mg kg$^{-1}$ effectively cured the study mice, with no detectable parasites in the liver smears, whereas control mice dosed with vehicle alone showed a high level of infection (*Figure 2*). A second in vivo study with mice dosed twice-daily at 30, 10 or 3 mg kg$^{-1}$ suppressed infection in the murine model by 99.5%, 63.5% and 16.0%, respectively, establishing a dose-dependent anti-leishmanial effect within the range of 3–50 mg kg$^{-1}$. These results give an estimated $ED_{50}$ and $ED_{90}$ of 7.3 and 21.5 mg kg$^{-1}$, respectively (*Figure 2—figure supplement 1*). At 30 and 50 mg kg$^{-1}$ delamanid compares favourably with miltefosine (98.8–99.8% suppression at 30 mg kg$^{-1}$), which exemplifies the therapeutic potential of delamanid.

A third in vivo study with a further reduced delamanid dose of 1 mg kg$^{-1}$ resulted in a suppression of parasitaemia of 86.3% compared with control mice, proving unexpectedly superior to dosing at 3 or 10 mg kg$^{-1}$ (*Figure 2*). A subsequent experiment encompassing a range of doses (10, 3, 1 mg kg$^{-1}$, 5 days) in a single study showed a similar hormetic effect, with twice daily dosing at 1 mg kg$^{-1}$ being more efficacious than 10 mg kg$^{-1}$. However, this study also demonstrated that there is some variability in the efficacy of delamanid at lower doses (*Figure 2—source data 1*).

The hormetic effect was also observed in an extended dosing experiment in which delamanid was instead dosed twice-daily for 10 days at 10, 3 or 1 mg kg$^{-1}$, with the suppression of infection being 92.3%, 24.3% and >99.9%, respectively. A second 10-day experiment with a broader range of doses (30, 10, 3, 1, 0.3 mg kg$^{-1}$) further confirmed the hormetic effect. In addition, this study demonstrated that further reducing the delamanid dose (0.3 mg kg$^{-1}$) resulted in a reduction in efficacy comparable to dosing at 3 mg kg$^{-1}$, resulting in a biphasic dose response relationship (*Figure 2*).

## Blood levels of orally dosed delamanid in a mouse model

It is important to understand the pharmacokinetic and pharmacodynamic (PK/PD) behaviour of delamanid in order to optimise the efficacious dosing regimen (*Velkov et al., 2013*). By measuring the change in drug concentration over time in *L. donovani*-infected mice, two standard PK parameters can be obtained: maximum concentration ($C_{max}$) in blood; and the area under the curve (AUC), a measure of total drug exposure over time. The drug concentration over time is measured in order to determine whether the concentration of a drug exceeds the minimum inhibitory concentration (MIC, $EC_{90}$ in this case) and, if so, for how long (time over MIC, $T_{>MIC}$). Parameters such as $C_{max}$/MIC, AUC/MIC and $T_{>MIC}$ are important for achieving drug efficacy in an in vivo model of disease. Both $C_{max}$ and AUC measure the total drug level in blood or plasma; however, only unbound drug

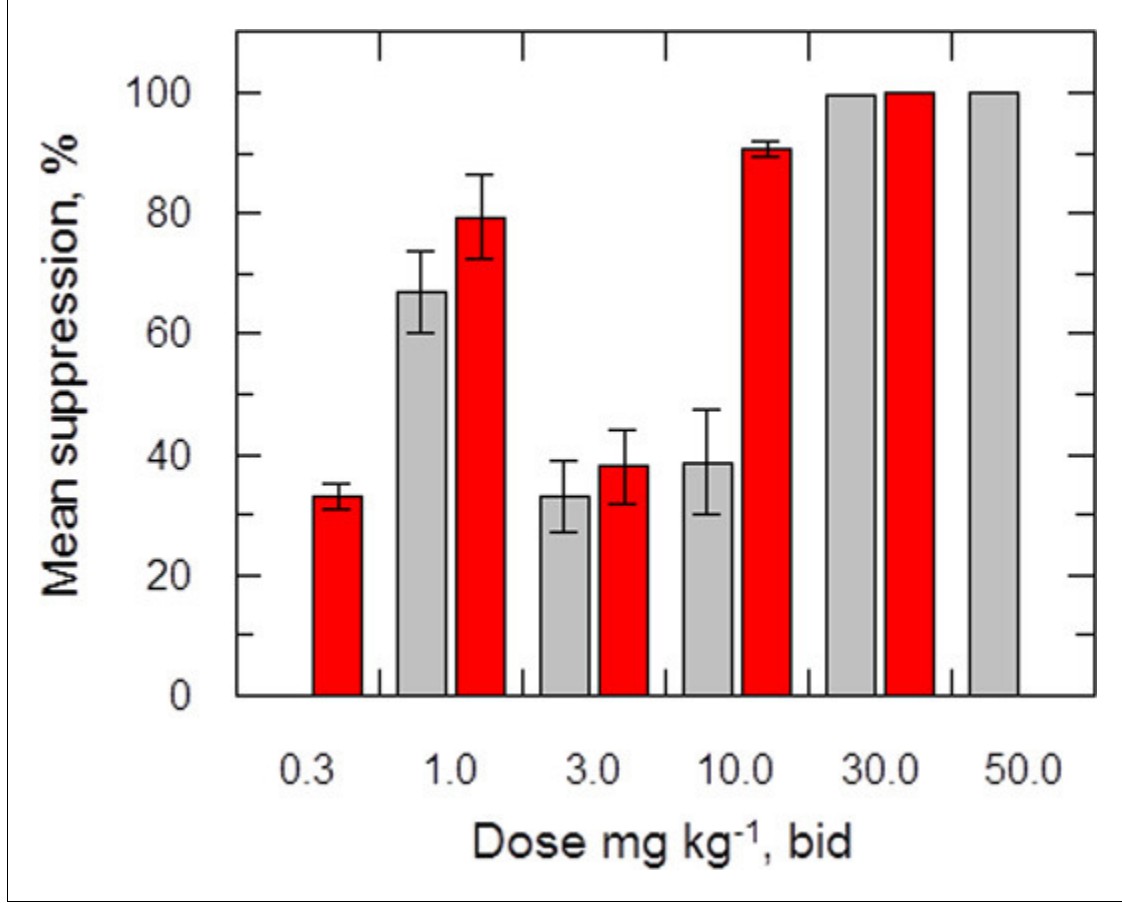

**Figure 2.** Effects of drug treatment on the parasite burden of mice infected with *L. donovani*. Groups of mice (five per group) infected with *L. donovani* (strain LV9) were dosed with drug vehicle (orally), miltefosine (orally) or delamanid (twice daily, orally) on day 7 post-infection and for a total of 5 or 10 days. Two days after the final dose, animals were humanely euthanized and parasite burdens were determined microscopically by examining Giemsa-stained liver smears. Grey bars, 5-day treatment; red bars, 10 day treatment. This graph shows the combined data from six individual animal studies; n = 9, or 10 for 1, 3 and 10 mg/kg dosing; for all other experiments n = 5. These data are available in tabular form in *Figure 2—source data 1*.

The following source data and figure supplement are available for figure 2:

**Source data 1.** Efficacy and PK/PD data from all experiments.

**Figure supplement 1.** $ED_{50}$ determination for delamanid in a mouse model of VL.

molecules are able to bind to their targets (*Bohnert and Gan, 2013*). Therefore, the plasma protein binding level (expressed as the fraction unbound, Fu) of delamanid was also measured and used to calculate an adjusted $EC_{90}$ (assay $EC_{90} \times 1/Fu$) for comparison with blood concentration over time.

Accordingly, the blood levels of the drug were measured at intervals (up to 8 hr post dose) during the in vivo efficacy studies. Data for the first and ninth dose in a 5-day twice daily treatment experiment (*Figure 3A,B*) show a dose-dependent response with accumulation over time. A similar effect was noted in a 10-day study (1st and 19th dose; *Figure 3C,D*). More detailed analysis of the combined PK data from five experiments (including two 10-day treatment studies) shows a linear relationship between doses of 0.3–10 mg $kg^{-1}$ and peak blood concentration ($C_{max}$) or area under the curve ($AUC_{(0-t)}$) with accumulation from day 1 through to day 10 (*Figure 3E,F*). The 10 and 30 mg $kg^{-1}$ doses should provide adequate coverage over the $EC_{90}$ (120 ng $ml^{-1}$) as measured for the *L. donovani* isolate LV9 in macrophages over a 3 day exposure (*Table 1*). However, due to high protein binding, the free fraction (Fu = 0.0045) cannot account for biological activity in vivo at any dose. An

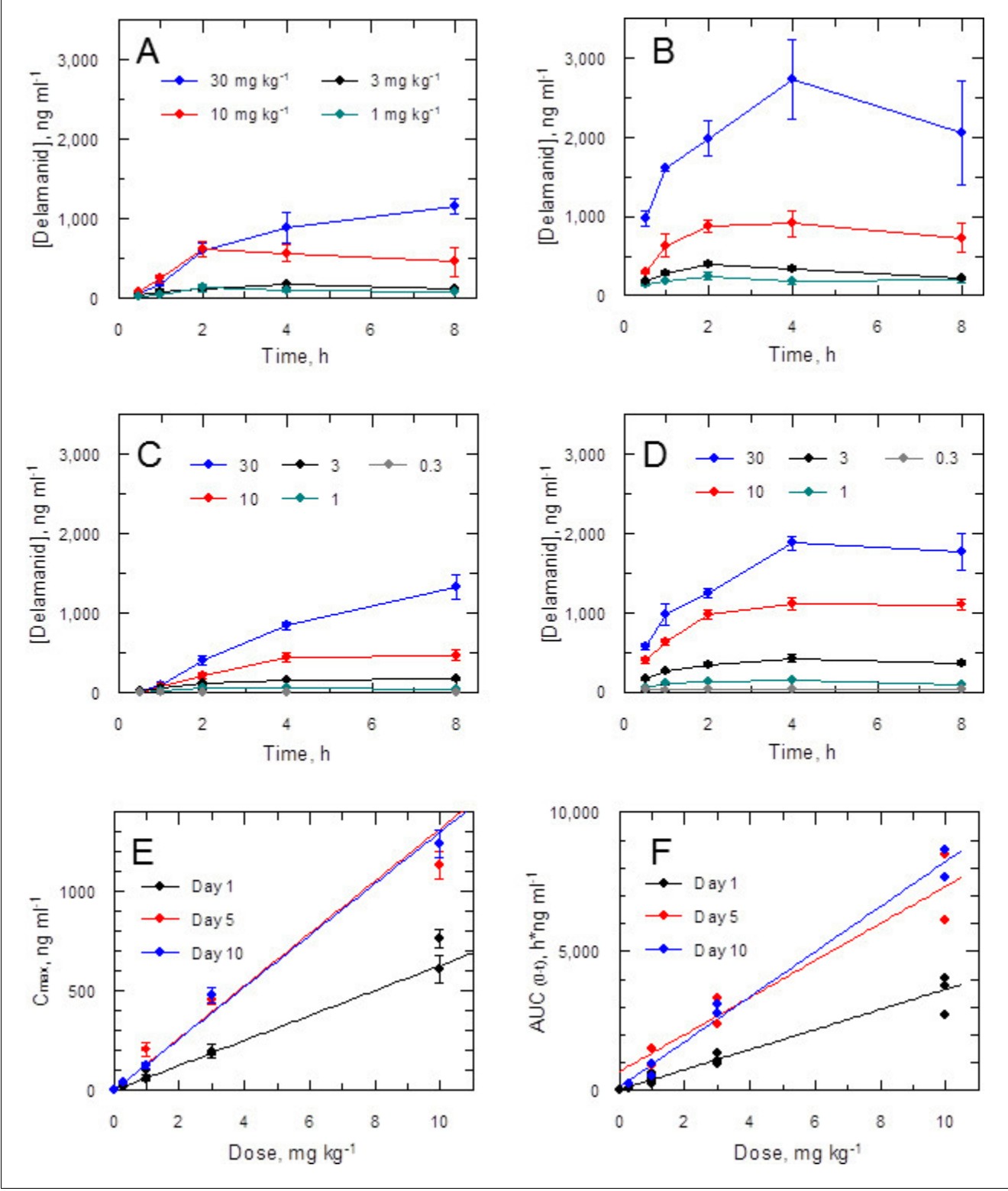

**Figure 3.** Pharmacokinetic behaviour of delamanid in infected mice. (**A** and **B**) show the blood levels of delamanid following the first oral dose on day 1 (**A**) and the penultimate oral dose on day 5 (**B**) for 1, 3, 10 and 30 mg kg$^{-1}$ b.i.d. (teal, black, red and blue symbols, respectively). Error bars are SEM (n = 3 for 30 mg kg$^{-1}$, n = 6 all other doses). (**C** and **D**) show the blood levels of delamanid from a single VL PK/PD study following the first oral dose on day 1 (**C**) and the penultimate oral dose on day 10 (**D**) for 0.3, 1, 3, 10 and 30 mg kg$^{-1}$ b.i.d. (grey, teal, black, red and blue symbols, respectively). Error bars are SEM (n = 5). (**E** and **F**) show the relationship between dose with $C_{max}$ or AUC$_{(0-8\ h)}$, respectively, after the first oral dose on day 1 (black), or the

*Figure 3 continued on next page*

*Figure 3 continued*

penultimate dose on day 5 (red) or day 10 (blue). Error bars in (E) are SEM (n ranges from 3–8 depending upon day and dose – see Figure 2 – source data1). Lines in (E) and (F) are best fits by linear regression.

explanation for why the free drug theory (*Bohnert and Gan, 2013*) is not applicable in this case is presented below.

## Delamanid-mediated cell killing

To determine whether delamanid was cytostatic or cytotoxic, mid-log promastigotes were incubated with drug concentrations equivalent to 10 times the $EC_{50}$ value (*Figure 4A*). Growth of drug-treated cultures ceased almost immediately with cell numbers declining after 8 hr and no live parasites visible at 24 hr. To determine the actual point where treated cells lost viability, at defined intervals parasites were washed and sub-cultured without drug. No viable parasites could be recovered after 12 hr in the presence of drug, confirming that delamanid is rapidly leishmanicidal. In support of this apparent rapid mechanism of cell killing, $EC_{50}$ values determined after 24, 48 and 72 hr were essentially identical (*Figure 4B*). In addition, the potency ($EC_{50}$ value) of delamanid was found to be dependent on the initial cell density (*Figure 4C*) and on the assay serum concentration (*Figure 4D*).

## Delamanid – mode of action studies

Many nitroheterocyclics require bio-activation of their nitro groups to become biologically active. In *Mycobacterium tuberculosis*, delamanid is assumed to be reductively activated by the same unusual deazaflavin (F420)-dependent nitroreductase (Ddn) known to activate the closely related nitroimidazo-oxazine drug PA-824 (*Manjunatha et al., 2006*; *Singh et al., 2008*; *Manjunatha et al., 2009*). In the absence of a Ddn homologue in *Leishmania*, we assessed whether the reduction of delamanid is catalysed by the NADH-dependent bacterial-like nitroreductase (NTR) already shown to activate the nitroimidazoles fexinidazole and nifurtimox in these parasites (*Wyllie et al., 2012*). The potency of delamanid was determined against parasites overexpressing NTR. Increased concentrations of NTR in these transgenic parasites were confirmed by a 13-fold increase in their sensitivity to nifurtimox ($EC_{50}$ of $8.0 \pm 0.2$ and $0.61 \pm 0.006$ µM for WT and transgenic parasites, respectively *Figure 5A*), known to undergo two-electron reduction by NTR (*Hall et al., 2011*). However, overexpression of NTR in promastigotes did not significantly alter their sensitivity to delamanid ($EC_{50}$ of $4.5 \pm 0.004$ and $4.1 \pm 0.003$ nM for WT and transgenic parasites, respectively) (*Figure 5B*). To confirm that the same was also true in the amastigote stage of these parasites, metacyclic promastigotes overexpressing NTR were used to infect mouse peritoneal macrophages. The resulting intracellular parasites were found to be just as sensitive to delamanid as WT parasites with $EC_{50}$ values of $57.8 \pm 2.1$ and $55.2 \pm 4.3$ nM, respectively (*Figure 5C*). These findings indicate that NTR does not play a role in the activation of delamanid in *L. donovani* in either stage of the life cycle and that the mechanism of action of this nitroheterocyclic drug is different from that of fexinidazole.

## Metabolism of delamanid in *L. donovani*

Given that NTR does not activate delamanid in *L. donovani* promastigotes and the requirement of the nitro group for biological activity, it was important to determine if the drug is metabolised in culture. To address this issue, the concentration of delamanid was determined by UPLC-MS/MS in cultures of promastigotes over a 24 hr period. Delamanid is known to be primarily metabolised in plasma by albumin (*Shimokawa et al., 2015*) and to a lesser extent by CYP3A4, CYP1A1, CYP2D6 and CYP2E1 (*Sasahara et al., 2015*). Thus, the concentration of delamanid in culture medium without parasites was measured over the same time period as a control. In the presence of medium alone, delamanid decreased linearly in a concentration-dependent manner (*Figure 6A*). However, in the presence of *L. donovani* promastigotes the rate of disappearance of delamanid was markedly increased, such that the drug had essentially disappeared by 6 hr (*Figure 6B*). The net amount of delamanid metabolised by parasites as a function of time is also linear and dependent on the initial concentration in the medium (*Figure 6C*). Linear regression of these data revealed that the rate of cell metabolism is not saturated up to the top concentration tested (*Figure 6D*). Analogous experiments using mouse peritoneal macrophages and THP-1 monocytes found no evidence of delamanid

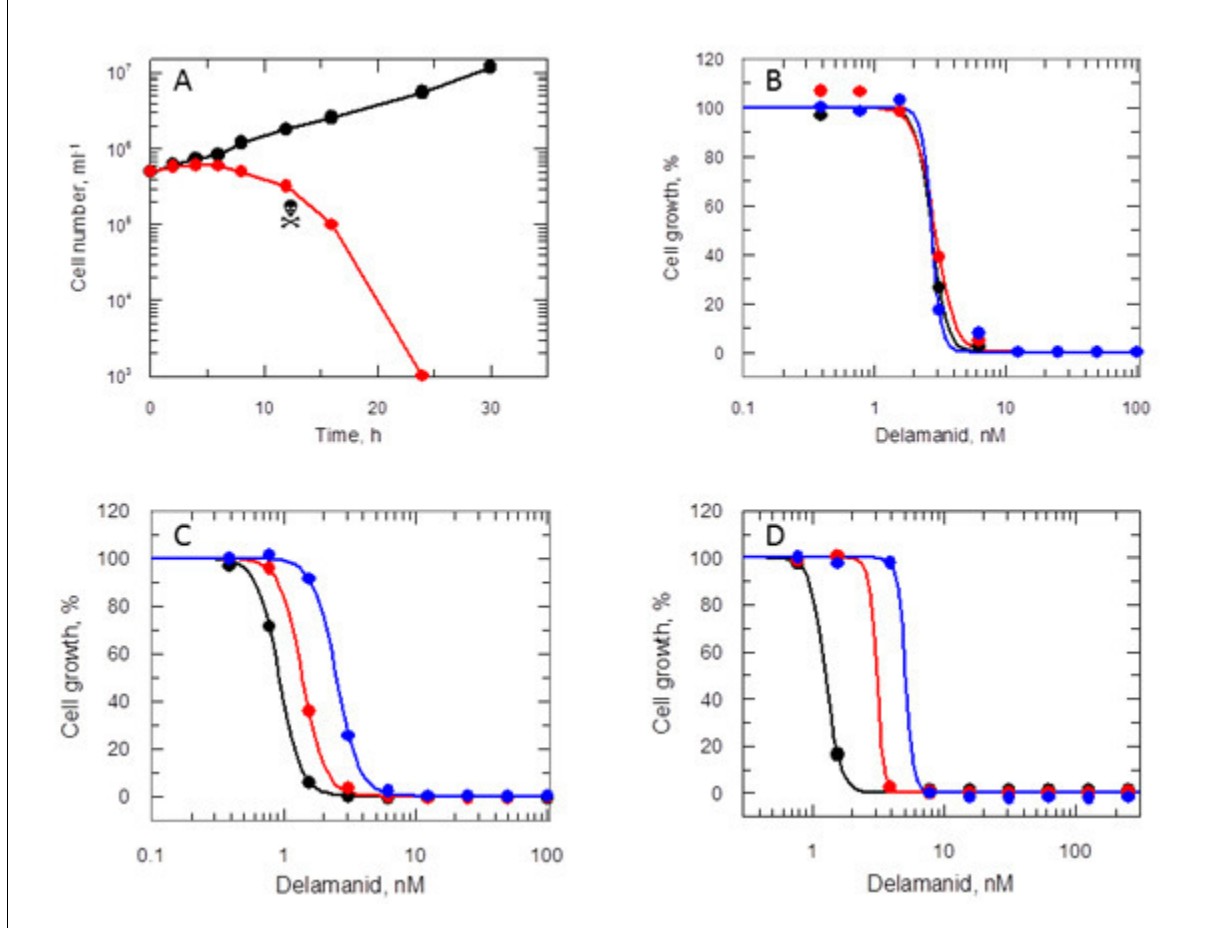

**Figure 4.** Effects of delamanid on L. donovani promastigotes. (**A**) Delamanid causes rapid cell killing. Promastigotes were exposed to delamanid (10 times $EC_{50}$) and samples removed at intervals to determine cell density and cell viability. Black symbols: no inhibitor; red symbols: plus drug; ☠ the point of irreversible drug toxicity. (**B**) Drug sensitivity is independent of exposure beyond 24 hr. Black, red and blue symbols are $EC_{50}$ curves determined after 24, 48 and 72 hr, respectively. (**C**) Drug sensitivity is cell-density dependent. Black, red and blue symbols are $EC_{50}$ curves determined after 72 hr, with initial seeding densities of $10^3$, $10^4$ and $10^5$ cells $ml^{-1}$, respectively. (**D**) Drug sensitivity is serum dependent. Black, red and blue symbols are $EC_{50}$ curves determined after 72 hr in the presence of 5, 10 and 20% FCS, respectively.

metabolism by these host cell lines. Elucidation of the chemical identity of the delamanid metabolite (s), their possible role in parasite killing and the enzyme(s) responsible for their biosynthesis will be the focus of future studies.

## Discussion

For diseases of poverty such as visceral leishmaniasis there is limited financial incentive to initiate expensive, high risk and time-consuming *de novo* drug discovery programmes. Consequently, the repurposing of existing drugs has become an attractive approach towards the identification of much needed new treatments for VL and other neglected parasitic diseases (*Andrews et al., 2014*; *Wyllie et al., 2012*). The recently approved anti-tubercular drug delamanid (*Ryan and Lo, 2014*) was deemed to be of particular interest as a number of other nitroimidazoles have been shown to also possess promising anti-leishmanial activities (*Wyllie et al., 2012*; *Patterson et al., 2013*; *Gupta et al., 2015*).

In the current study, we show that delamanid is highly active in vitro against intracellular *L. dono-vani* amastigotes ($EC_{50}$ 0.087 μM) with activity superior to that of both the current VL drug miltefo-sine ($EC_{50}$ 3.3 μM) and the active sulfone metabolite of the VL clinical candidate fexinidazole ($EC_{50}$ 5.3 μM) in the same assay (*Wyllie et al., 2012*). The in vitro anti-leishmanial activity of delamanid

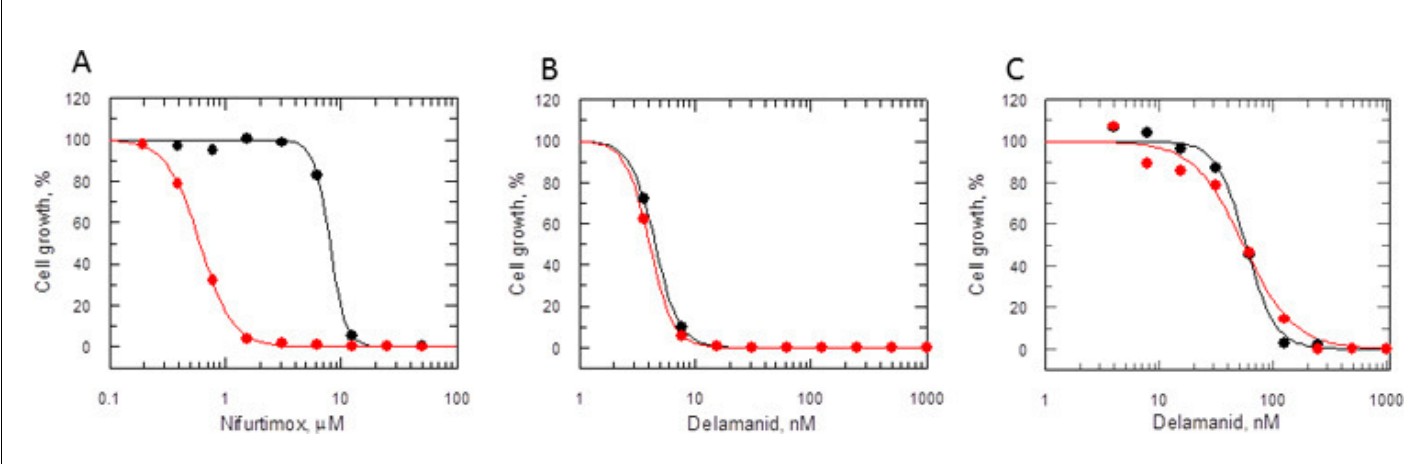

**Figure 5.** Mode of action of delamanid is distinct from nifurtimox. (A, B and C) Susceptibility to nifurtimox (A) is increased in NTR-overexpressing promastigotes (red symbols), but not to delamanid (B) compared to WT cells (black symbols). Data are the mean of triplicate cultures from a single experiment. (C) The susceptibility of delamanid is not increased in NTR-overexpressing amastigotes in macrophages (red symbols) compared to WT (black symbols). Data are the mean of duplicate cultures from a single experiment.

shows the same enantiomeric specificity as the delamanid analogue DNDI-VL-2098 in vivo (*Gupta et al., 2015*). Similarly, the *R*-enantiomer of an analogue of delamanid has been shown to be more potent against *M. tuberculosis* than its corresponding *S*-enantiomer (*Sasaki et al., 2006*). However, this contrasts with the closely related nitroimidazole PA-824, where the enantiomeric specificity for *L. donovani* and *M. tuberculosis* is opposite (*Patterson et al., 2013*). Delamanid is rapidly leishmanicidal with a cell-density dependent potency and as expected, demonstrated no observable toxicity in a mammalian cell assay. The observed shift in drug sensitivity with increasing serum concentration is likely due to increased metabolism by albumin to inactive metabolites, rather than changes in free drug concentration due to protein binding.

Delamanid is orally bioavailable, well tolerated, shows dose linearity up to 10 mg kg$^{-1}$ and accumulation with repeated administration in agreement with previous pharmacokinetic (PK) studies in mice (*Matsumoto et al., 2006*). However, the pharmacokinetic / pharmacodynamic (PK/PD) relationship is not straightforward.

First, the PK/PD relationship does not fit with the free drug hypothesis, which states that, in the absence of energy-dependent transport processes, the extracellular and intracellular free drug concentrations are equal after steady-state equilibrium has been achieved, and that only the free drug is able to bind to the target to exert its pharmacological effect (*Bohnert and Gan, 2013*; *Smith et al., 2010*). Based on the in vitro intra–macrophage amastigote EC$_{90}$ value (120 ng ml$^{-1}$), the high PPB of delamanid (F$_u$ 0.0045) results in an adjusted intra macrophage *L. donovani* EC$_{90}$ of 26,700 ng ml$^{-1}$, a concentration that is not achieved in whole blood at any point in the efficacy study. Therefore, the efficacy of delamanid in the VL animal model is unexpected. However, there are exceptions to the free drug theory, such as drugs that form active metabolites resulting in inactivation (covalent or otherwise) of multiple targets (*Smith et al., 2010*). The parasite-specific metabolism presented here is entirely consistent with this exception.

Second, the pronounced bi-phasic suppression of parasite burden at high and low doses of delamanid in vivo is highly unusual. This U-shaped dose-response curve is reminiscent of hormesis in toxicology (*Calabrese and Baldwin, 2003*; *2002*), but, to our knowledge, is unprecedented in microbiology. This effect is not observed in dose response curves in vitro, so must be related to a physiological or metabolic threshold response in the infected drug-treated animal. In vivo delamanid is proposed to undergo primary metabolism with loss of the nitro group, mainly catalysed by albumin (*Committee for Medicinal Products for Human Use, 2013*; *Shimokawa et al., 2015*). Further metabolism to seven other metabolites is thought to occur via hydrolysis reactions and oxidation by CYP3A4 (*Committee for Medicinal Products for Human Use, 2013*; *Sasahara et al., 2015*). The cause of this U-shaped dose-response curve is currently not known. One possibility is that a drug

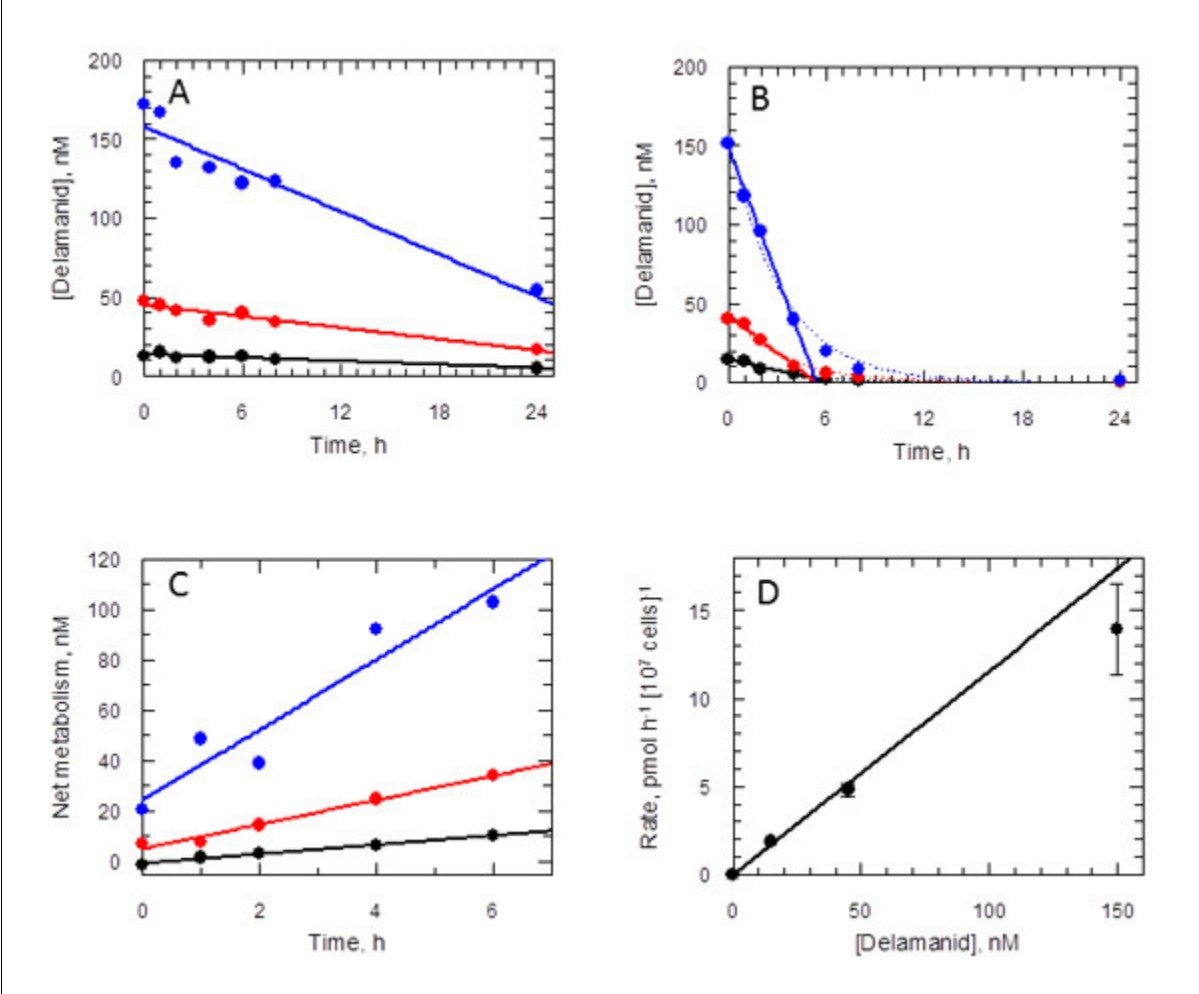

**Figure 6.** Delamanid metabolism in L. donovani promastigotes. (**A**) Medium plus delamanid alone and (**B**) cells incubated in medium plus delamanid. Delamanid concentrations added are 15, 45 and 150 nM (black, red and blue, respectively). The lines represent best fits by linear regression for all data points in (**A**) and 0 to 5 hr in (**B**). The dotted line in (**B**) is the best fit by non-linear regression to a single exponential decay. (**C**) Net metabolism of delamanid by cells was obtained by subtraction of (**A**) from (**B**). Data fitted by linear regression gave correlation coefficients of 0.996, 0.991 and 0.951 for delamanid concentrations of 15, 45 and 150 nM, respectively. (**D**) Rates of delamanid metabolism obtained from (**C**) are linear up to 150 nM (correlation coefficient 0.996, explicit errors used in fit).

metabolite of delamanid antagonises the bio-activation of delamanid, or antagonises the downstream effects in the leishmania parasite. The formation of a putative antagonist metabolite would have to show a saturable sigmoidal dose response, such that at higher concentrations, delamanid, or an active parasite-specific metabolite thereof, are able to displace the antagonist from the bio-activating enzyme, or proteins related to the downstream effect respectively. It should be noted that in the same VL animal model the related nitroimidazole (*S*)-PA-824 was also more efficacious at a lower dose (30 vs 100 mg kg$^{-1}$) (*Patterson et al., 2013*). Further studies with (*S*)-PA-824 should be conducted to determine if this compound also displays a hormetic PK/PD relationship and establish if this is a chemotype-related characteristic.

Plots of $C_{max}$ versus parasite suppression and calculated AUC$_{(0-24 \text{ hr})}$ versus parasite suppression (*Figure 7A,B*) suggest that the delamanid blood levels required for cure in the VL model exceed those observed in TB patients receiving the drug. Increasing the dosing duration in the in vivo VL model from 5 to 10 days improved the mean parasite suppression at all investigated doses (*Figure 2*) and resulted in some mice with no detectable liver parasites when dosed at 1 mg kg$^{-1}$ (*Figure 2—source data 1*). As it known that delamanid is tolerated for up to six months (*Committee for*

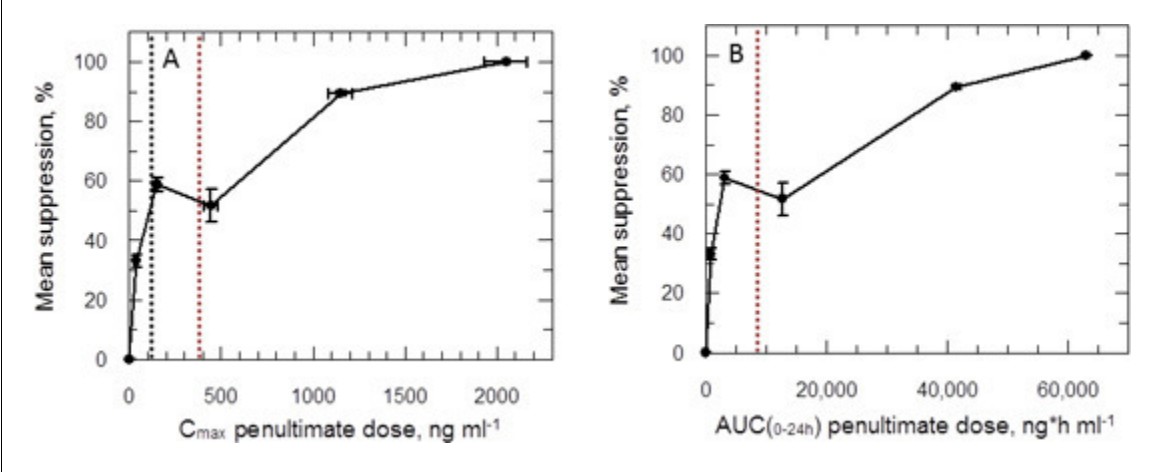

**Figure 7.** PK/PD relationships in mice. (**A** and **B**) Mean suppression of parasite burden as a function of $C_{max}$ for the penultimate dose (panel **A**) and extrapolated $AUC_{(0–24\ hr)}$ for the last day of the 10-day treatment regimen (panel **B**). The black dotted line in (**A**) is the $EC_{90}$ value obtained for infected macrophages after 72 hr exposure (120 ng ml$^{-1}$). The red dotted line in (**A**) and (**B**) represents the mean delamanid $C_{max}$ (375–400 ng ml$^{-1}$) and mean $AUC_{(0–24\ hr)}$ (7000–8000 h*ng ml$^{-1}$) obtained in 144 TB patients after 14 days treatment with 100 mg, oral, once daily from day 14–56 (*Sasahara et al., 2015*). The data in this graph were derived from a single in vivo study; related aggregated data from previous studies is shown in *Figure 7—figure supplement 1*.

The following figure supplement is available for figure 7:

**Figure supplement 1.** (**A** and **B**) Mean suppression of parasite burden as a function of $C_{max}$ for the penultimate dose and extrapolated $AUC_{(0–24\ hr)}$ for the last day of the 5- and 10-day treatment regimens, respectively.

*Medicinal Products for Human Use, 2013*) further extending the duration of the VL model beyond 10 days should be considered. The VL target product profile calls for a treatment regimen of <10 days. However, in the current VL therapy, miltefosine is dosed orally for 28 days, so extended dosing should be clinically acceptable. Note, the study mice dosed twice-daily at 1 mg kg$^{-1}$ have drug levels lower than that achieved in human TB patients dosed once daily at 100 mg. Given that this low dose is more efficacious than any other dose below 30 mg kg$^{-1}$, model studies of extended duration should focus around this dosing level.

Despite a wealth of pharmacokinetic data in patients and human volunteers (*Committee for Medicinal Products for Human Use, 2013*) the unusual PK/PD relationship hinders our ability to accurately predict the outcome of delamanid dosing in VL patients. Indeed, careful examination of parasite suppression versus $C_{max}$ (*Figure 7—figure supplement 1*) shows that the mean $C_{max}$ in delamanid-treated TB patients corresponds to an efficacy minimum in the VL model. Given that delamanid is rapidly metabolised by leishmania-infected macrophages in vitro, we examined the effect of delamanid with a 5 day exposure with daily drug and medium change. This gave an $EC_{50}$ value of 28.0 ± 1.6 nM (slope 2.7) from which an $EC_{90}$ could be calculated (62.7 nM or 33.5 ng ml$^{-1}$). This concentration is lower than the lowest observed $C_{max}$ in the efficacy studies (*Figure 7A*). Thus, careful design of the dosing regimen for VL patients may avoid the risk that treatment will lack efficacy due to reaching a $C_{max}$ and $AUC_{(0–24\ hr)}$ within the higher ineffective concentration range.

The nature of the parasite-specific metabolising / activating enzyme(s) is not known, but is clearly distinct from the deazaflavin-dependent nitroreductase in *M. tuberculosis* (*Manjunatha et al., 2006*) and the nitroreductase in leishmania involved in the activation of fexinidazole metabolites (*Wyllie et al., 2012*). The identification of this target, and the metabolites that it produces, are the focus of our current work. The cell-density dependent potency of delamanid is consistent with the formation of a putative reactive, covalent metabolite. In addition, the rapidly cytocidal activity of delamanid is consistent with the rapid rate of drug metabolism by *L. donovani* in culture. In terms of drug development the divergent modes of action for fexinidazole and delamanid are advantageous, as the likelihood of cross-resistance developing is reduced, and the potential for their co-administration as a combination therapy is retained.

The current practice in the pharmaceutical industry is to avoid developing compounds containing a nitro-aromatic group due to the known liabilities of this class, particularly potential mutagenicity and carcinogenicity. As a result, outside the anti-infectives there are relatively few nitro-aromatic drugs. Indeed, nitro-aromatic moieties are relatively common in drugs or clinical candidates for kinetoplastid diseases compared to chemotherapies for other indications. This over-representation is linked to the mechanism of action of nitro-drugs; selective bio-activation by parasitic bacterial-like NTRs leading to selective anti-parasitic activity. The studies presented herein are consistent with delamanid also being activated by a parasite-specific enzyme absent from host cells. Preclinical studies have demonstrated that delamanid is not mutagenic (*Matsumoto et al., 2006*). Moreover, repeated oral administration in mice or rats for up to 104 weeks showed no evidence of carcinogenicity (*Committee for Medicinal Products for Human Use, 2013*). Taken together these points alleviate some of the concerns ordinarily associated with the development of nitro-drugs, although long term safety can only be established after extensive clinical use in relevant populations.

While revising this manuscript, Thompson and co-workers reported the structure-activity relationships for an extensive series of bicyclic nitro-compounds, including delamanid (*Thompson et al., 2016*). These authors observed partial efficacy in *L. infantum*-infected hamsters dosed with delamanid, not inconsistent with our findings. The lower efficacy observed by these authors could be due to one or more of the following: inadequate dosing (once versus twice per day); different animal model (hamster versus mouse); or different species of leishmania (*L. infantum* is less sensitive than *L. donovani*). Although this paper found VL-2098 to have superior efficacy in a once daily treatment regimen, pre-clinical development of this compound has been abandoned due to testicular toxicity (http://www.dndi.org/diseases-projects/portfolio/completed-projects/vl-2098/ last accessed 3$^{rd}$ April 2016). Importantly, this reproductive toxicity has not been observed with delamanid (*Committee for Medicinal Products for Human Use, 2013*).

Delamanid meets many of the criteria specified in the target product profile (TPP) for VL [http://www.dndi.org]. It is rapidly cytocidal and thus potentially efficacious in immunocompromised patients such as those co-infected with HIV (*Alvar et al., 2008*) particularly since delamanid is not associated with any clinically relevant anti-retroviral drug-drug interactions (*Ryan and Lo, 2014*; *Blair and Scott, 2015*). Due to the prevalence of TB-VL coinfection in Ethiopia (*Hurissa et al., 2010*) and Sudan (*El-Safi et al., 2004*) the TPP also specifies that any new treatments should be compatible with TB medications, a stipulation met by delamanid (*Blair and Scott, 2015*). Delamanid can be administered orally, an important requirement for patients who have limited access to even the most basic of health care facilities, whereas liposomal amphotericin B (AmBisome) has to be administered intravenously and requires cold storage for stability. Perhaps the most challenging issue may be cost of goods. Assuming an efficacious dose of 100 mg once-daily for in excess of 10 days and the market cost of delamanid in developed countries (US$42 per 50 mg tablet) (*Lessem, 2014*), the predicted cost per patient would be over US$840, significantly higher than that specified in the VL TPP (<US$10 or <$80 per course) and predicted to be more expensive than other current treatment strategies (*Meheus et al., 2010*). However, differential pricing or financial arrangements may reduce the cost for this neglected disease of poverty. Additional factors also support the development of delamanid as a VL therapy. Importantly, the results presented here demonstrate that delamanid is active against a number of clinically-relevant field isolates. In addition, we are currently investigating the potential of delamanid to be repurposed for Chagas' disease.

In summary, these data suggest that delamanid has the potential to be repurposed as a VL therapy. Additional VL animal model studies exploring the effect of extended delamanid dosing beyond 10 days should be investigated.

## Materials and methods

### Cell lines and culture conditions

All leishmania strains follow the WHO International Code designating the animal from which the parasite was isolated, country, date of isolation and strain designation (see International Leishmania Network http://leishnet.net/site/?q=node/5). The WHO designations and origins of each Leishmania isolate used are detailed in *Table 2*. The clonal *Leishmania donovani* cell line LdBOB was grown as promastigotes at 24°C in 10% FCS, as previously described (*Goyard et al., 2003*), except when

**Table 2.** *Leishmania* isolates used in delamanid drug sensitivity studies.

| Species | WHO Code | Laboratory Code | Origin | Year of Isolation | Provided by |
|---|---|---|---|---|---|
| *L. donovani* | MHOM/SD/62/1S CL2D [a] | LdBOB | Sudan | 1962 | via Professor Stephen Beverley, Washington University |
| *L. donovani* | MHOM/ET/67/HU3 | LV9 | Ethiopia | 1967 | via Professor Jennie Blackwell, Cambridge University |
| *L. donovani* | MHOM/IN/02/BHU1 | BHU1 | India | 2002 | via LSHTM, London, UK |
| *L. donovani* | MHOM/SU/09/SUKA001 | SUKA001 | Sudan | 2009 | via LSHTM, London, UK |
| *L. donovani* | MHOM/IN/80/DD8 | DD8 | India | 1980 | via LSHTM, London UK |
| *L. infantum* | MHOM/MA/67/ITMAP263 | ITMAP263 | Morocco | 1967 | via LSHTM, London, UK |
| *L. major* | MHOM/IL/81/Friedlin | Friedlin | Israel | 1981 | via LSHTM, London, UK |

[a] Derived from this strain (*Goyard et al., 2003*)

investigating the effect of serum concentration on drug efficacy, in which case 5, 10, or 20% FCS was used. Transgenic LdBOB promastigotes expressing the *L. major* nitroreductase (LmjF.05.0660) enzyme (*Wyllie et al., 2012*) were cultured under identical conditions in the presence of nourseothricin (100 µg ml$^{-1}$). *L. major* promastigotes (Friedlin strain) were grown in M199 medium (Caisson Laboratories, Logan, UT) with supplements, as previously described (*Oza et al., 2005*). *L. donovani* (LV9 strain) ex vivo amastigotes were used in both in vitro and in vivo drug sensitivity assays. Amastigotes were derived from hamster spleens, as previously described (*Wyllie and Fairlamb, 2006*). All other *Leishmania* clinical isolates (*Table 2*) were grown in RPMI 1640 (Sigma, UK) supplemented with 20% FCS, 100 µM adenine, 5 µM hemin, 20 mM MES, 3 µM 6-biopterin and 1 mM biotin. In all cases the FCS used was certified as mycoplasma free.

## Chemical synthesis of delamanid and analogues

Delamanid was prepared as previously described (*Sasaki et al., 2006*; *Kiyokawa and Aki, 2005*) (*Figure 1—figure supplement 1*). Modification of the delamanid synthetic route afforded (*S*)-delamanid and des-nitro-delamanid (*Figure 1—figure supplements 1–2*). Compound purity was determined by liquid chromatography-mass spectrometry, with all compounds found to be >95% pure. For in vivo experiments, delamanid was further analysed by ultra high-performance liquid chromatography-mass spectrometry (UPLC-MS), with all batches found to be of >98% purity. The optical rotation of delamanid was in close agreement to the published value (*Sasaki et al., 2006*), confirming the optical purity of the material used in this study. Detailed synthetic procedures and analysis of key compounds and intermediates are provided in Appendix 1.

## In vitro drug sensitivity assays against promastigotes

To examine the effects of test compounds on growth, triplicate cultures were seeded with $1 \times 10^5$ parasites ml$^{-1}$. Parasites were grown in the presence of drug for 72 hr, after which 50 µM resazurin was added to each well and fluorescence (excitation of 528 nm and emission of 590 nm) measured after a further 2 hr incubation (*Jones et al., 2010*). Data were processed using GRAFIT (version 5.0.13; Erithacus software) and fitted to a 2-parameter equation, where the data are corrected for background fluorescence, to obtain the effective concentration inhibiting growth by 50% (EC$_{50}$):

$$ y = \frac{100}{1 + \left( \frac{[I]}{EC_{50}} \right)^m} $$

In this equation [I] represents inhibitor concentration and *m* is the slope factor. Experiments were repeated at least two times and the data is presented as the weighted mean plus weighted standard deviation (*Young, 1962*). When investigating the speed of drug-mediated cell killing, parasites were grown in the presence of drug for 24, 48, or 72 hr in an otherwise identical assay. The same assay was used to investigate the effect of seeding density upon drug efficacy, except that the number of parasites used to seed the assays was varied to be either $10^3$, $10^4$ or $10^5$ parasites ml$^{-1}$.

### Cytocidal effects of delamanid on *L. donovani* promastigotes

Delamanid was added to early-log cultures of LdBOB promastigotes ($\sim 1 \times 10^6$ ml$^{-1}$) at concentrations equivalent to 10 times its EC$_{50}$ value. At intervals, the cell density was determined, samples of culture (500 µl) removed, washed and resuspended in fresh culture medium in the absence of drug. The viability of drug-treated parasites was monitored for up to 24 hr and the point of irreversible drug toxicity determined by microscopic examination of subcultures after 5 days.

### In vitro drug sensitivity assays in mouse macrophages and toxicity to HepG2 cells

In-macrophage drug sensitivity assays were carried out using starch-elicited mouse peritoneal macrophages and hamster-derived ex vivo amastigotes (*Wyllie et al., 2012*) or metacyclic promastigotes (*Wyllie et al., 2013*), where appropriate. Assays to determine the sensitivity of HepG2 cells to test compounds were carried out precisely as previously described (*Patterson et al., 2013*). HepG2 were obtained from ATCC and routinely tested for mycoplasma contamination by Mycoplasma Experience Ltd.

### In vitro pharmacokinetic and biophysical properties

The PPB of delamanid was determined by the equilibrium dialysis method (*Jones et al., 2010*). The aqueous solubility of delamanid was measured using a laser nephelometry-based method (*Patterson et al., 2013*).

### In vivo drug sensitivity

Groups of female BALB/c mice (5 per group) were inoculated intravenously (tail vein) with approximately $2 \times 10^7$ *L. donovani* LV9 amastigotes harvested from the spleen of an infected hamster (*Wyllie and Fairlamb, 2006*). From day 7 post-infection, groups of mice were treated with either drug vehicle only (orally), with miltefosine (30 mg kg$^{-1}$ orally), or with delamanid (1, 3, 10, 30 or 50 mg kg$^{-1}$ orally). Miltefosine was administered once daily for 5, or 10 days, with vehicle and delamanid administered twice daily over the same period. Drug dosing solutions were freshly prepared each day, and the vehicle for delamanid was 0.5% hydroxypropylmethylcellulose, 0.4% Tween 80, 0.5% benzyl alcohol, and 98.6% deionized water. On day 14 (for 5 day dosing experiments), or day 19 post-infection (for 10 day dosing experiments), all animals were humanely euthanized and parasite burdens were determined by counting the number of amastigotes/500 liver cells (*Wyllie et al., 2012*). Parasite burden is expressed in Leishman Donovan Units (LDU): mean number of amastigotes per 500 liver cells × mg weight of liver (*Bradley and Kirkley, 1977*). The LDU of drug-treated samples are compared to that of untreated samples and the percent inhibition calculated. ED$_{50}$ values were determined using GRAFIT (version 5.0.13; Erithacus software) by fitting data to a 2-parameter equation, as described above.

### Determination of delamanid exposure in infected mice after oral dosing

Blood samples (10 µl) from 3 of 5 infected mice (see in vivo drug sensitivity above) in each dosing group were collected from the tail vein and placed into Micronic tubes (Micronic BV) containing deionized water (20 µl). Samples were taken following the first dose on the first (day 7 post-infection) and last day of dosing (day 11, or 16 post-infection) at 0.5, 1, 2, 4 and 8 hr post-dose. Diluted blood samples were freeze-thawed three times prior to bioanalysis. The concentration of delamanid in mouse blood was determined by UPLC-MS/MS on a Xevo TQ-S (Waters, UK) by modification of that described previously for the analysis of fexinidazole (*Sokolova et al., 2010*) and PK parameters determined using PKsolutions software (Summit, USA). AUC$_{(0–24\ hr)}$ was extrapolated from the calculated AUC$_{(0-8\ hr)}$, with second daily dose administered at 8 hr post first daily dose.

### Rate of delamanid metabolism in *L. donovani* promastigotes

Rate of metabolism studies were carried out at 15, 45 and 150 nM delamanid (equivalent to 1-, 3- and 10-times EC$_{50}$) in culture medium alone and in the presence of wild type *L. donovani* promastigotes ($1 \times 10^7$ parasites ml$^{-1}$). At 0, 0.5, 1, 2, 4, 6, 8 and 24 hr aliquots were removed, precipitated by addition of a 3-fold volume of acetonitrile and centrifuged (1665 × g, 10 min, room temperature).

The supernatant was diluted with water to maintain a final solvent concentration of 50% and stored at −20°C prior to UPLC-MS/MS analysis, as described below.

UPLC-MS/MS was performed on a Waters Acquity UPLC interfaced to a Xevo TQ-S MS. Chromatographic resolution was achieved on a 2.1 × 50 mm Acquity BEH C18, 1.7 µm column which was maintained at 40°C with an injection volume of 8 µl. The mobile phase consisted of A: deionized water plus 0.01% (v/v) formic acid and B: acetonitrile plus 0.01% (v/v) formic acid at a flow rate of 0.6 ml min$^{-1}$. The initial gradient was 5% B held for 0.5 min before increasing to 95% B from 0.5–2 min, where it was held from 2–2.6 min before decreasing back to 5% B from 2.6–3 min. Mass spectra were obtained using electrospray ionization (ESI), in positive ion mode with the following conditions: capillary 3.5 kV; desolvation temperature 600°C; source temperature 150°C; desolvation gas flow (nitrogen) 1000 l h$^{-1}$ and collision gas (argon) gas of 0.15 ml min$^{-1}$. Multiple reaction monitoring (MRM) was performed for delamanid using the transition 535.02 > 351.80 at a cone voltage of 16 V and collision energy of 33 V. Data was processed using the TargetLynx feature of Mass Lynx v4.1.

## Acknowledgements

We thank Gina MacKay for performing high-resolution mass spectrometry (MS) analyses and for assistance with performing nuclear magnetic resonance and other MS analyses, Liam Ferguson for VL model efficacy studies, and the biology and DMPK teams of the Drug Discovery Unit at the University of Dundee for HepG2 cell screening and in vitro pharmacokinetic assays, respectively. This work was funded by grants from the Wellcome Trust (079838 and 092340).

## Additional information

### Funding

| Funder | Grant reference number | Author |
| --- | --- | --- |
| Wellcome Trust | 079838 | Alan H Fairlamb |
| Wellcome Trust | 092340 | Alan H Fairlamb |

The funders had no role in study design, data collection and interpretation, or the decision to submit the work for publication.

### Author contributions

SP, Design and synthesis of chemical compounds, Acquisition of data, Analysis and interpretation of data, Drafting or revising the article; SW, Drug sensitivity assays against leishmania, Construction of transgenic parasites, Acquisition of data, Analysis and interpretation of data, Drafting or revising the article; SN, Delamanid metabolism in leishmania parasites, Acquisition of data, Analysis and interpretation of data, Drafting or revising the article; LS, FRCS, In vivo animal studies, Acquisition of data, Analysis and interpretation of data, Drafting or revising the article; JLA, Chemical synthesis, Acquisition of data, Analysis and interpretation of data, Drafting or revising the article; MO-C, Pharmacokinetic studies, Acquisition of data, Analysis and interpretation of data, Drafting or revising the article; KDR, Drug metabolism and pharmacokinetic studies, Conception and design, Analysis and interpretation of data, Drafting or revising the article; AHF, Research concept and project management, Analysis and interpretation of data, Drafting or revising the article

### Author ORCIDs

Susan Wyllie, http://orcid.org/0000-0001-8810-5605
Alan H Fairlamb, http://orcid.org/0000-0001-5134-0329

### Ethics

Animal experimentation: All animal experiments were approved by the Ethical Review Committee at the University of Dundee and performed under the Animals (Scientific Procedures) Act 1986 (UK Home Office Project Licence PPL 70/8274) in accordance with the European Communities Council Directive (86/609/EEC).

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

Appendix 1

## Procedures for the synthesis of delamanid and analogues

### General

Chemicals and solvents were purchased from Sigma-Aldrich (UK), Alfa Aesar (UK), Apollo Scientific (UK), Fisher Chemicals (UK), Tokyo Chemical Industry (UK) and VWR (UK) and were used as received. Air and moisture sensitive reactions were carried out under an inert atmosphere of nitrogen. Analytical thin-layer chromatography (TLC) was performed using pre-coated TLC plates (layer 0.20 mm silica gel 60 with fluorescent indicator UV254, from Merck, UK). Developed plates were air-dried and analysed under a UV lamp (UV254/365 nm), and/or with chemical stains where appropriate. Flash column chromatography was performed using prepacked silica gel cartridges (230-400 mesh, 35-70 µm, from Teledyne ISCO) using a Teledyne ISCO CombiFlash Rf. $^1$H-NMR, $^{13}$C-NMR, $^{19}$F-NMR, and 2D-NMR spectra were recorded on a Bruker Avance DPX 500 spectrometer ($^1$H at 500.1 MHz, $^{13}$C at 125.8 MHz, $^{19}$F at 470.5 MHz), or a Bruker Avance III HD ($^1$H at 400.1 MHz, $^{13}$C at 100.6 MHz,). Chemical shifts (δ) are expressed in ppm recorded using the residual solvent as the internal reference in all cases. Signal splitting patterns are described as singlet (s), doublet (d), triplet (t), quartet (q), multiplet (m), broad (br), or a combination thereof. Coupling constants ($J$) are quoted to the nearest 0.5 Hz. LC-MS analyses were performed with either an Agilent HPLC 1100 series connected to a Bruker Daltonics MicrOTOF or an Agilent Technologies 1200 series HPLC connected to an Agilent Technologies 6130 quadrupole LC/MS, where both instruments were connected to an Agilent diode array detector. LCMS chromatographic separations were conducted with either a Waters XBridge C18 column, 50 mm × 2.1 mm, 3.5 µm particle size, or Waters XSelect C18 column, 30 mm × 2.1 mm, 2.5 µm particle size; mobile phase, water/ acetonitrile +0.1% HCOOH, or water/acetonitrile +0.1% NH$_3$. High-resolution electrospray measurements were performed on a Bruker Daltonics MicrOTOF mass spectrometer. Preparative HPLC separations were performed with a Gilson HPLC (321 pumps, 819 injection module, 215 liquid handler/injector) connected to a Gilson 155 UV/vis detector. HPLC chromatographic separations were conducted using a Waters XBridge C18 column, 19 × 100 mm, 5 µm particle size; mobile phase, water/acetonitrile +0.1% NH$_3$, or HCOOH. Optical rotation measurements were performed using a PerkinElmer model 343 polarimeter.

### Synthesis of delamanid (7)

Delamanid was synthesised according to published procedures (*Figure 1—figure supplement 1*). In brief, commercially available (2$R$)-2-methylglycidyl-4-nitrobenzoate (**1**) (Sigma-Aldrich) was transformed to ($R$)-2-bromo-1-((2-methyloxiran-2-yl)methyl)-4-nitro-1$H$-imidazole (**5**) in four steps via intermediates **2**, **3** and **4** as described by Sasaki and coworkers (*Sasaki et al., 2006*). Epoxide **5** was subsequently reacted with 4-(4-(4-(trifluoromethoxy)phenoxy)piperidin-1-yl) phenol (**6**) (*Sasaki et al., 2006*; *Kiyokawa and Aki, 2005*) and sodium hydride as described below to furnish delamanid (**7**).

Synthesis of ($R$)-2-methyl-6-nitro-2-((4-(4-(4-(trifluoromethoxy)phenoxy)piperidin-1-yl) phenoxy) methyl)-2,3-dihydroimidazo[2,1-$b$]oxazole (delamanid, OPC-67683, **7**) (*Sasaki et al., 2006*).

Solid NaH (60% suspension in oil, 19 mg, 0.48 mmol) was added to a solution of phenol **6** (141 mg, 0.40 mmol) and epoxide **5** (126 mg, 0.48 mmol) in anhydrous DMF (5 mL) at 0°C. The reaction was then allowed to warm to room temperature and subsequently heated to 50°C for 1.5 hr. Upon completion the reaction mixture was added to satd. aq. NaCl:EtOAc (1:1, 50 mL), the layers separated and the aq. layer extracted with EtOAc (3×25 mL). The combined EtOAc layers were then dried over MgSO$_4$, filtered and the solvent removed under reduced pressure. The crude product was purified by column chromatography (24 g silica, 0:100→100:0 EtOAc:

hexane) to give the title compound as a pale yellow solid (104 mg, 49% yield). $R_f$ (silica, 50:50 EtOAc:hexane) 0.24. $^1$H-NMR (500 MHz, CDCl$_3$) δ 7.55 (s, 1H, ArH), 7.16-7.13 (m, 2H, AA′BB′, 2×ArH), 6.93-6.89 (m, 4H, 2×AA′BB′, 4×ArH), 6.80-6.77 (m, 2H, AA′BB′, 2×ArH), 4.50 (d, 1H, $J$=10 Hz, C$H$H), 4.44-4.40 (m, 1H, CH), 4.18 (d, 1H, $J$=10 Hz, CH$H$), 4.06-4.02 (m, 2H, 2×C$H$H), 3.40-3.35 (m, 2H, 2×CH$H$), 3.03-2.98 (m, 2H, 2×C$H$H), 2.13-2.08 (m, 2H, 2×C$H$H), 1.98-1.92 (m, 2H, 2×CH$H$), 1.77 (s, 3H, CH$_3$). $^{19}$F-NMR (470 MHz, CDCl$_3$) δ -58.4. $^{13}$C-NMR (125 MHz, CDCl$_3$) δ 156.0 (C), 155.8 (C), 151.7 (C), 147.3 (C), 146.9 (C), 142.8 (C), 122.5 (CH), 120.6 (q, $J$=204 Hz, CF$_3$), 118.6 (CH), 116.8 (CH), 115.9 (CH), 115.6 (CH), 112.5 (CH), 93.2 (C), 72.7 (CH), 72.3 (CH$_2$), 51.5 (CH$_2$), 47.9 (CH$_2$), 30.6 (CH$_2$), 23.2 (CH$_3$). MS (ES+): $m/z$ (%) 535 (100) [M+H]$^+$. HRMS (ES+): calcd. for C$_{25}$H$_{26}$F$_3$N$_4$O$_6$ [M+H]$^+$ 535.1799, found 535.1779 (3.7 ppm). $[\alpha]_D^{20} = -10.8 (c\ 1.02,\ \text{CHCL}_3)$.

## Synthesis of (S)-delamanid (13)

The synthesis of (S)-delamanid (**13**) was accomplished via a modification of the published (*Sasaki et al., 2006*) route towards delamanid (*Figure 1—figure supplement 1*). In brief, commercially available (2S)-2-methylglycidyl-4-nitrobenzoate (**8**) (Sigma-Aldrich) was transformed to (S)-2-bromo-1-((2-methyloxiran-2-yl)methyl)-4-nitro-1H-imidazole (**12**) in four steps via intermediates **9**, **10** and **11** as described (*Sasaki et al., 2006*). Epoxide **12** was subsequently reacted with 4-(4-(4-(trifluoromethoxy)phenoxy)piperidin-1-yl)phenol (**6**) (*Sasaki et al., 2006*) and sodium hydride as described below to afford (S)-delamanid (**13**).

Synthesis of (S)-2-methyl-6-nitro-2-((4-(4-(4-(trifluoromethoxy)phenoxy)piperidin-1-yl) phenoxy) methyl)-2,3-dihydroimidazo[2,1-b]oxazole ((S)-delamanid, **13**) (*Sasaki et al., 2006*).

Solid NaH (60% suspension in oil, 5.4 mg, 0.14 mmol) was added to a solution of phenol **6** (40 mg, 0.11 mmol) and epoxide **5** (36 mg, 0.14 mmol) in anhydrous DMF (2.5 mL) at 0°C. The reaction was then allowed to warm to room temperature and subsequently heated to 50°C for 1.5 hr. Upon completion water (0.2 mL) was added to the reaction, the resultant mixture was then filtered and purified by reverse phase HPLC (20:80→95:5 MeCN:water +0.1% HCOOH) to give the title compound as a pale yellow solid (22 mg, 36% yield). $^1$H-NMR (500 MHz, CDCl$_3$) δ 7.55 (s, 1H, ArH), 7.15-7.13 (m, 2H, AA′BB′, 2×ArH), 6.93-6.89 (m, 4H, 2×AA′BB′, 4×ArH), 6.80-6.76 (m, 2H, AA′BB′, 2×ArH), 4.49 (d, 1H, $J$=10.5 Hz, C$H$H), 4.44-4.39 (m, 1H, CH), 4.18 (d, 1H, $J$=10.5 Hz, CH$H$), 4.05-4.00 (m, 2H, 2×C$H$H), 3.39-3.35 (m, 2H, 2×CH$H$), 3.03-2.98 (m, 2H, 2×C$H$H), 2.13-2.07 (m, 2H, 2×C$H$H), 1.98-1.91 (m, 2H, 2×CH$H$), 1.77 (s, 3H, CH$_3$). MS (ES+): $m/z$ (%) 535 (100) [M+H]$^+$. HRMS (ES+): calcd. for C$_{25}$H$_{26}$F$_3$N$_4$O$_6$ [M+H]$^+$ 535.1799, found 535.1754 (8.4 ppm).

## Synthesis of des-nitro-delamanid (18)

The synthesis of des-nitro-delamanid (**18**) was accomplished via a modification of the published (*Sasaki et al., 2006*) route towards delamanid (*Figure 1—figure supplement 2*). Full details of the synthetic route are given below.

## Synthesis of (R)-2-hydroxy-2-methyl-3-(2-nitro-1H-imidazol-1-yl)propyl 4-nitrobenzoate (14)

Neat DIPEA (465 mg, 3.6 mmol) was added to a suspension of 2-nitroimidazole (1.02 g, 9.0 mmol) and epoxide **1** (2.13 g, 9.0 mmol) in anhydrous EtOAc (45 mL) and stirred at 65°C for 20 hr. The reaction was subsequently diluted with MeCN (15 mL) and the reaction temperature increased to 77°C, after which the resultant solution was stirred for an additional 24 hr. The reaction mixture was then directly purified by column chromatography (120 g silica,

10:90→100:0 EtOAc:heptane) to give the title compound as a white solid (2.27 g, 72% yield). $R_f$ (silica, 50:50 EtOAc:hexane) 0.16. [1]H-NMR (400 MHz, DMSO-*d6*) δ 8.38-8.35 (m, 2H, AA′BB′, 2×ArH), 8.26-8.23 (m, 2H, AA′BB′, 2×ArH), 7.59 (s, 1H, ArH), 7.17 (s, 1H, ArH), 5.49 (s, 1H, OH), 4.67 (d, 1H, *J*=14.0 Hz, C*H*H), 4.60 (d, 1H, *J*=14.0 Hz, CH*H*), 4.13 (s, 2H, $CH_2$), 1.14 (s, 3H, $CH_3$). [13]C-NMR (100 MHz, DMSO-*d6*) δ 164.5 (C), 150.9 (C), 146.3 (C), 135.3 (C), 131.3 (CH), 128.7 (CH), 127.4 (CH), 124.3 (CH), 70.6 (C), 69.9 ($CH_2$), 54.8 ($CH_2$), 22.8 ($CH_3$). MS (ES+): *m/z* (%) 351 (100) [M+H]$^+$, 701 (40) [2M+H]$^+$.

## Synthesis of (*R*)-2-methyl-3-(2-nitro-1H-imidazol-1-yl) propane-1,2-diol (15)

Solid $K_2CO_3$ (18.6 mg, 0.135 mmol) was added to a solution of ester **14** (944 mg, 2.70 mmol) in anhydrous MeOH (25 mL) and stirred at room temperature for 20 hr. A solution of HCl (6 N, aq, 0.55 mL) and solid $MgSO_4$ (550 mg) were then added to the reaction and the resultant mixture stirred for 1 hr, before being filtered through a plug of celite, and the solvent removed under reduced pressure. The crude product was purified by column chromatography (80 g silica, 0:50:50→20:80:0 EtOH:EtOAc:heptane) to give the title compound as a pale yellow solid (475 mg, 83% yield). $R_f$ (silica, EtOAc) 0.33. [1]H-NMR (400 MHz, DMSO-*d6*) δ 7.50 (s, 1H, ArH), 7.12 (s, 1H, ArH), 4.96 (t, 1H, *J*=5.0 Hz, OH), 4.79 (br s, 1H, OH), 4.48 (d, 1H, *J*=14.0 Hz, C*H*H), 4.39 (d, 1H, *J*=14.0 Hz, CH*H*), 3.14 (dd, 1H, *J*=11.0, 5.0 Hz, C*H*H), 3.07 (dd, 1H, *J*=11.0, 5.0 Hz, CH*H*), 0.95 (s, 3H, $CH_3$). [13]C-NMR (100 MHz, DMSO-*d6*) δ 146.5 (C), 128.4 (CH), 127.0 (CH), 72.1 ($CH_2$), 67.3 (C), 54.8 ($CH_2$), 22.7 ($CH_3$). MS (ES+): *m/z* (%) 202 (100) [M+H]$^+$.

## Synthesis of (*R*)-2-hydroxy-2-methyl-3-(2-nitro-1H-imidazol-1-yl)propyl methane sulfonate (16)

Neat MsCl (344 mg, 3.0 mmol) was added to a solution of diol **3** (402 mg, 2.0 mmol) and anhydrous pyridine (791 mg, 10.0 mmol) in $CH_2Cl_2$ (20 mL) at 0°C before being allowed to warm to room temperature and stirred for an additional 16 hr. The reaction was then poured onto HCl (1N, aq, 20 mL) and the pH of the aqueous layer adjusted to 2.5. The layers were subsequently separated and the aq, extracted with $CH_2Cl_2$ (4 × 50 mL). The combined $CH_2Cl_2$ layers were dried over $MgSO_4$, filtered and the solvent removed under reduced pressure to give the crude product as a yellow oil which was reacted on without further purification, or analysis.

## Synthesis of (*R*)-1-((2-methyloxiran-2-yl)methyl)-2-nitro-1*H*-imidazole (17)

Neat DBU (335 mg, 2.2 mmol) was added to a solution of crude mesylate **16** (~2.0 mmol) in anhydrous EtOAc (20 mL) and stirred at room temperature for 16 hr. The reaction was subsequently washed with satd. aq. NaCl (50 mL), followed by extraction of the aq. layer with EtOAc (2×50 mL). The combined EtOAc layers were dried over $MgSO_4$, filtered, and the solvent removed under reduced pressure. The crude product was purified by column chromatography (80 g silica, 25:75→100:0 EtOAc:heptane) to give the title compound as a thick clear oil (58 mg, 15% yield over 2 steps). $R_f$ (silica, EtOAc) 0.6. [1]H-NMR (400 MHz, CDCl$_3$) δ 7.16-7.13 (m, 2H, 2×ArH), 4.93 (d, 1H, *J*=14.5 Hz, C*H*H), 4.41 (d, 1H, *J*=14.5 Hz, CH*H*), 2.69 (d, 1H, *J*=4.0 Hz, C*H*H), 2.45 (d, 1H, *J*=4.0 Hz, CH*H*), 1.36 (s, 3H, $CH_3$). [13]C-NMR (100 MHz, CDCl$_3$) δ 145.0 (C), 128.4 (CH), 126.7 (CH), 55.6 (C), 53.1 ($CH_2$), 51.6 ($CH_2$), 18.8 ($CH_3$). MS (ES+): *m/z* (%) 184 (100) [M+H]$^+$.

## Synthesis of des-nitro-delamanid ((R)-2-methyl-2-((4-(4-(4-(trifluoromethoxy)phenoxy) piperidin-1-yl)phenoxy) methyl)-2,3-dihydroimidazo[2,1-b]oxazole) (18)

Solid NaH (60% dispersion in oil) (30 mg, 0.76 mmol) was added to a solution of epoxide **17** (58 mg, 0.32 mmol) and phenol **6** (134 mg, 0.38 mmol) in anhydrous DMF (4 mL) at 0°C. The reaction was then allowed to warm to room temperature followed by heating at 50°C for 5 hr. The reaction mixture was subsequently added to satd. aq. NaCl:EtOAc (1:1, 20 mL), the layers separated and the aq. layer extracted with EtOAc (3×10 mL). The combined EtOAc layers were then dried over $MgSO_4$, filtered and the solvent removed under reduced pressure. The crude product was purified by reverse phase HPLC (5:95→95:5 MeCN:water + 0.1% $NH_3$) to give the title compound as a clear semi-solid (51 mg, 30% yield). $^1$H-NMR (500 MHz, $CDCl_3$) δ 7.16-7.13 (m, 2H, AA'BB', 2×ArH), 6.94-6.90 (m, 4H, 4×ArH), 6.84-6.81 (m, 2H, AA'BB', 2×ArH), 6.71 (d, 1H, $J$=1.5 Hz, ArH), 6.57 (d, 1H, $J$=1.5 Hz, ArH), 4.44-4.40 (m, 1H, CH), 4.29 (d, 1H, $J$=9.5 Hz, C$H$H), 4.12 (d, 1H, $J$=9.5 Hz, C$H$H), 4.02 (d, 1H, $J$=9.5 Hz, C$H$H), 3.86 (d, 1H, $J$=9.5 Hz, C$H$H), 3.40-3.35 (m, 2H, 2×C$H$H), 3.03-2.98 (m, 2H, 2×C$H$H), 2.14-2.09 (m, 2H, 2×C$H$H), 1.99-1.92 (m, 2H, 2×C$H$H), 1.73 (s, 3H, $CH_3$). $^{19}$F-NMR (470 MHz, $CDCl_3$) δ -58.4 ($CF_3$). $^{13}$C-NMR (100 MHz, $CDCl_3$) δ 159.1 (C), 155.9 (C), 152.3 (C), 146.6 (C), 142.8 (C), 129.5 (CH), 122.5 (CH), 118.6 (CH), 116.9 (CH), 115.4 (CH), 110.6 (CH), 92.1 (C), 72.8 (CH), 72.1 ($CH_2$), 51.4 ($CH_2$), 48.0 ($CH_2$), 30.6 ($CH_2$), 23.5 ($CH_2$). Note, the $CF_3$ resonance was not observed. MS (ES+): $m/z$ (%) 490 (80) [M+H]$^+$, 479 (100) [2M+H]$^+$. HRMS (ES+): calcd for $C_{25}H_{27}F_3N_3O_4$ [M+H]$^+$ 490.1948, found 490.1967 (-3.8 ppm).

