## [Decision Letter]

Thank you for submitting your work entitled "The anti-tubercular drug delamanid as a potential oral treatment for visceral leishmaniasis" for peer review at *eLife*. Your submission has been favorably evaluated by Prabhat Jha as the Senior editor, Jon Clardy as Reviewing Editor, and two reviewers including Clifton Barry.

The reviewers have discussed the reviews with one another and the Reviewing editor has drafted this decision to help you prepare a revised submission.

Summary:

Visceral leishmaniasis is a disease caused by protozoal parasites that imposes a significant burden of mortality and morbidity on impoverished tropical populations. All current therapies have significant liabilities. The authors have adopted a repurposing strategy to screen drugs for other indications for their potential to treat leishmaniasis, and through that approach they identified delamanid as a potential oral treatment. The manuscript addresses pharmacokinetic/pharmacodynamic (PK/PD) properties of delamanid and concludes that it could be a useful therapeutic agent.

Essential revisions:

All reviewers noted the importance of improved treatments for visceral leishmaniasis, appreciated the benefits of a drug-repurposing strategy, and enthused about the potency of delamanid. They also recommended publication with revisions focusing on the following points:

1) The authors and reviewers are experienced in drug development, but almost all *eLife* readers are not. It would be useful in revising the manuscript to keep this in mind and provide clues to what the various tests are trying to determine. For example, most readers will be unfamiliar with a term like 'area under the curve (AUC)’ in pharmacology, and adding a brief note that it measures the total dose of drug and is useful for comparing different formulations would help enormously. It's a challenge, but a challenge worth addressing. The report as written is informative to a specialist and almost unintelligible to anyone else.

2) It would be useful to add some clarification about different developmental stages of the parasite: the difference between a promastigote and amastigote, delamanid's efficacy against different stages, possibility of host metabolism of delamanid altering efficacy, and the possibility of different targets for different stages.

3) The presence of a nitro group, which is common in this field and quite familiar to the authors, should be addressed for the general reader. Why are nitro groups relatively rare in other drugs but (almost) indispensable in drugs for these parasites? What have the authors done to address concerns about the presence of nitro groups? A very informative review is referenced, but again just a few sentences would help.

4) The hormetic or U-shaped dose response curve is somewhat unusual, and the addition of blood concentration in Figure 3 might help. It's also possible that this issue could be left undiscussed as it doesn't seem to have any ready interpretation that would help the reader or provide any insights that would illuminate the main points of the article.

*Reviewer #2:*

This manuscript presents compelling data to support the conclusion that the tuberculosis drug delamanid-can be repurposed as an inhibitor of *Leishmania donovani*. The compound is very effective in vitro versus *L. donovani* promastigotes. More importantly it is effective in vivo when given twice daily, orally, at 1 mg per kilogram for 10 days. I am not concerned with the U-shaped dose response curve or issues of drug cost. I agree with the authors that delamanid represents a potential oral treatment for visceral leishmaniasis but also conclude that the manuscript would be strengthened by attention to the following points:

1) The experiments illustrated in Figure 4 were carried out with *Leishmania* promastigotes. Did the authors also determine in vitro activity of the delamanid versus amastigotes within macrophages? Given the efficacy of this drug in the mouse model of infection one would expect efficacy in vitro as well versus intracellular amastigotes. The focus of this proposed experiment should be the effect of drug not against axenic amastigotes, but against amastigotes within macrophages.

2) Because metabolism of amastigotes differs from that of promastigotes, it would also be important to reevaluate whether the mechanism of action of delamanid is distinct from fexinidazole in amastigotes harvested from macrophages. This is important because the macrophage itself may modify the drug as well as the parasite.

3) A concern commonly voiced in the pharmaceutical industry with any nitro containing drug is the potential for genotoxicity and carcinogenesis. Was an Ames test carried out on delamanid? While this drug has been approved for use against tuberculosis, its potential for genotoxicity or cancer induction will not really be known until it has been used extensively in different populations "post market". While the TPP for leishmaniasis dictates a relatively short course of the drug, the length of treatment is only one variable and again, unfortunately, only extensive use clinically will reveal if delamanid is truly safe. The authors should at least comment on this possibility in the Discussion.

Also in this regard it should be noted that proponents of fexinidazole for the treatment of sleeping sickness have argued that although fexinidazole is positive in the Ames test, it is negative in mammalian genotoxic tests. They also argue that the nitroreductases necessary to produce free radicals from fexinidazole were not present in some bacteria. Nevertheless, one will not be certain that the positive Ames test is of the concern until fexinidazole has been used in several clinical settings. A major concern is the difference in the microbiome between a patient in India, Africa, and Europe. One cannot assume that the absence of an enzyme like a nitroreductase in one microbiome means it will be absent in all. Again this is not a criticism of the current manuscript, but at least a warning or recognition of this potential issue should be stated in the Discussion.

*Reviewer #3:*

The manuscript describes in vitro and in vivo activity against leishmania of a nitroimidazole-containing TB drug that has recently been EMA approved for the treatment of MDR tuberculosis. The most important and exciting finding is that delamanid is 30-50 times more active in vivo than either the current standard of care or one of the most promising clinical candidates for the treatment of VL. The authors have reported on analogs of this class of molecule before, with the rather disappointing finding that the other TB drug candidate in this class (PA-824) had only moderate activity. PA-824 is a chiral molecule with an S configuration on the oxazine ring; the surprising finding was that the opposite stereoisomer (which is not being developed for clinical use) was considerably more active. Additional screening revealed another molecule with similar stereospecificity that was similar to delamanid and the crucial insight in this work was that delamanid, (R)-PA-824, and the screening hit all presented similar stereochemical configurations suggesting delamanid might be potent against VL.

On balance the work is well done, carefully and thoughtfully presented and convincing that delamanid should be a strong candidate for repurposing for the treatment of VL – which is an extremely important bit of news in this disease.

There are a couple of things that could be improved upon particularly around the discussion of the "hormetic" dose-response curve which is very confusing and needs to be clarified – specifically I would suggest that they add the blood-concentration curve for the 1 mg kg^-1^dose to Figure 3 and show that this is not due to differences in accumulation rates for the lower dose. This is a really confusing piece of information and it is not clear if they have looked for this effect in vitro at the relevant concentration ranges in vitro to see if this is metabolism by the organism or metabolism by the host (if in fact their explanation of the antagonistic metabolite is correct).

There have been two (very) recent publications on the in vivo metabolism of delamanid (Shimokawa et al., Drug Metabol Dispos 2015 Aug;43(8)1277-83 and Sasahara et el in the same issue pp 1267-76) that should be cited and discussed. It seems at least plausible that the antagonistic metabolite is the molecule referred to as M1 in those publications that is formed by cleavage of the nitroimidazooxazole. They describe a simple looking procedure for producing M1 from human serum albumin that might allow the authors to more definitively address the plausibility of this pharmacodynamic mechanism.

My only other complaint is in the subsection “Delamanid – mode of action studies” where they discuss the lack of cross-resistance between fexinidazole resistant LV9 promastigotes and delamanid but state that the data will be published elsewhere. Either show the data or take this out of the Results section. Personally I don't think this is necessary as the lack of hypersensitivity of the NTR overexpressing parasite is already pretty convincing evidence that the drugs act by different mechanisms.

---

## [Author Response]

1) The authors and reviewers are experienced in drug development, but almost all eLife readers are not. It would be useful in revising the manuscript to keep this in mind and provide clues to what the various tests are trying to determine. For example, most readers will be unfamiliar with a term like 'area under the curve' (AUC) in pharmacology, and adding a brief note that it measures the total dose of drug and is useful for comparing different formulations would help enormously. It's a challenge, but a challenge worth addressing. The report as written is informative to a specialist and almost unintelligible to anyone else.

An explanatory paragraph to address this has been added to the beginning of the “Blood levels of orally dosed delamanid in a mouse model” Results section.

2) It would be useful to add some clarification about different developmental stages of the parasite: the difference between a promastigote and amastigote, delamanid's efficacy against different stages, possibility of host metabolism of delamanid altering efficacy, and the possibility of different targets for different stages.

A) Explanatory text has been added to the “in vitro sensitivity of *L. donovani* to (*S*)- and (*R*)-delamanid” section of the Results to clarify the difference between the developmental stages. The comparative efficacy against these stages is reported in Table 1.

B) We have carried out additional experiments on mouse macrophages and THP-1 cells and found no evidence for further metabolism by these host cell lines. This has been added to the text in the Results section “Metabolism of delamanid in L. donovani”.

C) The target enzyme(s) involved in activation of delamanid for each stage is not known. However, we have carried out an additional experiment (Figure 5) showing that overexpression of NTR does not affect the efficacy of delamanid against the intra macrophage amastigote. See Figure 5 and Results section “Delamanid – mode of action studies”. Thus NTR is not involved in either stage of the life cycle.

3) The presence of a nitro group, which is common in this field and quite familiar to the authors, should be addressed for the general reader. Why are nitro groups relatively rare in other drugs but (almost) indispensable in drugs for these parasites?

There are a number of potential explanations for the “over representation” of nitroaromatic-containing drugs/clinical candidates for kinetoplastid diseases relative to other indications: 1) as for other drugs against these diseases both nifurtimox and benznidazole were developed >40 years ago before there was a general (justifiable) aversion to the development of nitro drugs by the pharma industry; 2) the Drugs for Neglected Diseases initiative made a strategic decision to investigate the efficacy of nitroaromatic-containing compound collections against kinetoplastid parasites (leading to the development of fexinidazole and DNDI-VL-2098); and 3) the kinetoplastids possess a bacterial-like NTR which activates nitroaromatic prodrugs. Bacterial-like and mammalian-like NTRs have different substrate specificities; therefore, there is the potential for nitro drugs to be *selectively* toxic to parasites over mammalian cells.

A paragraph discussing this question is included in the Discussion (seventh paragraph).

What have the authors done to address concerns about the presence of nitro groups? A very informative review is referenced, but again just a few sentences would help.

We have not conducted any experiments of our own to specifically address concerns relating to the nitro functionality of delamanid. However, reference to the work of Matsumoto et al. is cited in the eighth paragraph of the Discussion, showing that there is no evidence for mutagenicity based on a comprehensive series of bacterial reverse mutation experiments.

4) The hormetic or U-shaped dose response curve is somewhat unusual, and the addition of blood concentration in Figure 3 might help.

See response to reviewer #3 below.

*It's also possible that this issue could be left undiscussed as it doesn't seem to have any ready interpretation that would help the reader or provide any insights that would illuminate the main points of the article.* Although we do not know why the U-shaped dose response curve is observed, we feel that it is so unusual that it merits some level of discussion, including hypotheses as to why it might occur. Therefore, we propose to leave the discussion as it is. Note, as our additional studies showed some variation in the efficacy at lower delamanid doses we have amended the abstract to reflect that doses of 30 mg/kg (5 days) are curative and those at 1 mg/kg (10 days) on average suppress parasite burden by 79%.

Reviewer #2:

1) The experiments illustrated in Figure 4 were carried out with Leishmania promastigotes. Did the authors also determine in vitro activity of the delamanid versus amastigotes within macrophages? Given the efficacy of this drug in the mouse model of infection one would expect efficacy in vitro as well versus intracellular amastigotes. The focus of this proposed experiment should be the effect of drug not against axenic amastigotes, but against amastigotes within macrophages.

Yes, see Table 1, showing delamanid is efficacious against intra macrophage amastigotes in vitro. The additional study displayed in Figure 5 using intracellular amastigotes overexpressing NTR shows that NTR does not activate delamanid in amastigotes either. For practical reasons the studies displayed in Figure 4 have not been repeated using intra macrophage amastigotes.

2) Because metabolism of amastigotes differs from that of promastigotes, it would also be important to reevaluate whether the mechanism of action of delamanid is distinct from fexinidazolein amastigotes harvested from macrophages. This is important because the macrophage itself may modify the drug as well as the parasite.

The new experiment displayed in Figure 5 (see point 1 above) shows that the mode of activation of delamanid is distinct from fexinidazole and nifurtimox in intracellular amastigotes.

We have established that the macrophage does not metabolise delamanid – see Essential revisions, point 2B above.

3) A concern commonly voiced in the pharmaceutical industry with any nitro containing drug is the potential for genotoxicity and carcinogenesis. Was an Ames test carried out on delamanid?

The results of a delamanid bacterial reverse mutagenesis test (Ames test) are reported in the literature, see Matsumoto et al.(2006) PloS Med. 3(11): e466. Delamanid (then called OPC-67683) was found not to be mutagenic. Prolonged (104 week) exposure of rats or mice to delamanid showed no evidence of carcinogenicity (EMA Assessment report 2013). These points have been added to the Discussion.

While this drug has been approved for use against tuberculosis, its potential for genotoxicity or cancer induction will not really be known until it has been used extensively in different populations "post market". While the TPP for leishmaniasis dictates a relatively short course of the drug, the length of treatment is only one variable and again, unfortunately, only extensive use clinically will reveal if delamanid is truly safe. The authors should at least comment on this possibility in the Discussion.

Text to cover the concerns associated with the potential mutagenicity of nitro drugs has been inserted in the Discussion (eighth paragraph).

Also in this regard it should be noted that proponents of fexinidazole for the treatment of sleeping sickness have argued that although fexinidazole is positive in the Ames test, it is negative in mammalian genotoxic tests. They also argue that the nitroreductases necessary to produce free radicals from fexinidazole were not present in some bacteria. Nevertheless, one will not be certain that the positive Ames test is of the concern until fexinidazole has been used in several clinical settings. A major concern is the difference in the microbiome between a patient in India, Africa, and Europe. One cannot assume that the absence of an enzyme like a nitroreductase in one microbiome means it will be absent in all. Again this is not a criticism of the current manuscript, but at least a warning or recognition of this potential issue should be stated in the Discussion.

Text has been inserted into the manuscript to highlight some of the known liabilities of nitro drugs (see above), specifically “although long term safety can only be established after extensive clinical use in relevant populations.”

Reviewer #3:

On balance the work is well done, carefully and thoughtfully presented and convincing that delamanid should be a strong candidate for repurposing for the treatment of VL – which is an extremely important bit of news in this disease.

*There are a couple of things that could be improved upon particularly around the discussion of the "hormetic" dose-response curve which is very confusing and needs to be clarified – specifically I would suggest that they add the blood-concentration curve for the 1* mg kg^-1^dose to Figure 3 and show that this is not due to differences in accumulation rates for the lower dose.

The blood concentration data for the 1 mg kg^-1^ b.i.d. 5 day study has been added to Figure 3 & B. In addition the delamanid blood concentration data for a new 0.3, 1, 3, 10 and 30 mg kg^-1^ b.i.d. 10 day study has been added (Figure 3). This data demonstrates that the U-shaped dose-response curve is not due to non-linear drug accumulation when dosed at 1 mg kg^-1^.

This is a really confusing piece of information and it is not clear if they have looked for this effect in vitro at the relevant concentration ranges in vitro to see if this is metabolism by the organism or metabolism by the host (if in fact their explanation of the antagonistic metabolite is correct).

We did not perform any specific experiments to look for the hormetic effect in vitro. However, as mentioned in the Discussion, analysis of the EC_50_ curves from assaying delamanid against promastigotes and intra macrophage amastigotes showed no hormetic effect at the drug concentration range used in the assays.

There have been two (very) recent publications on the in vivo metabolism of delamanid (Shimokawa et al., Drug Metabol Dispos 2015 Aug;43(8)1277-83 and Sasahara et el in the same issue pp 1267-76) that should be cited and discussed. It seems at least plausible that the antagonistic metabolite is the molecule referred to as M1 in those publications which is formed by cleavage of the nitroimidazooxazole. They describe a simple looking procedure for producing M1 from human serum albumin that might allow the authors to more definitively address the plausibility of this pharmacodynamic mechanism.

We thank the reviewer for highlighting the relevance of these recent publications to this study, and we have included citations to them in the manuscript. As per the reviewer’s suggestion, we do intend to investigate the effect of M1 on the metabolism of delamanid. However, the suggested experiments will probably take some time to complete, and so we intend to report them in an additional publication focussing on the metabolism of delamanid/mechanism of action, rather than be appended to the current manuscript.

My only other complaint is in the subsection “Delamanid – mode of action studies” where they discuss the lack of cross-resistance between fexinidazole resistant LV9 promastigotes and delamanid but state that the data will be published elsewhere. Either show the data or take this out of the Results section. Personally I don't think this is necessary as the lack of hypersensitivity of the NTR overexpressing parasite is already pretty convincing evidence that the drugs act by different mechanisms.

The following text has been removed as suggested, “In addition, in studies to be published elsewhere, LV9 promastigotes which had been selected for resistance to fexinidazole retained sensitivity to delamanid (3.24 ± 0.04 and 3.09 ± 0.03 nM for WT and resistant lines, respectively).”

Other alterations not suggested by the referees, or editor:

An additional affiliation has been added for some of the authors.

We have conducted two additional delamanid VL animal model studies since the original submission of this manuscript. The relevant efficacy and DMPK data from these studies has been aggregated with that included originally. This has resulted in small changes to the mean C_max_ and AUC values throughout, which is reflected by minor alterations to some figures.

The unusual U-shaped dose response is still visible despite changes in the mean parasite suppression. The only significant finding is that dosing at 1 mg kg^-1^ for 10 days is not curative in every instance (n has doubled from 5 to 10 mice) giving a new mean parasite suppression of 79%. Due to this observation, we have removed any writing which states that this dose is curative, and any resultant extrapolation as to what a curative human dose of delamanid might be. For example; 3 sentences have been removed from the Discussion; “However, plots of C_max_ versus […] observed in humans”.

Discussion, tenth paragraph. As we are no longer predicting that once daily 100 mg kg^-1^ dosing of delamanid will be efficacious, we have amended the treatment cost per patient to be >US$840.

Our summary statement at the end of the Discussion has been amended to reflect other changes in the Discussion section – see above.

As a substantial amount of data from six individual animal studies is now included in this manuscript we have produced a spreadsheet which shows detailed efficacy and DMPK data for each individual animal. This is to ensure the transparent communication of this large set of results. This spreadsheet is reported as a supplementary file.

Whilst this manuscript was under revision a paper describing the in vivo and in vitro activity of delamanid and analogues against *Leishmania spp*. has been published (Thompson et al. J Med Chem 2016). A paragraph has been added (Discussion, ninth paragraph to discuss the relevance of their results in relation to our own. We have also added an additional EC50 determination using the *L. infantum* strain used in the study of Thompson et al. to Table 1.

We have added the details of the software used to calculate AUC values to the manuscript. We apologise for failing to include this in the original submission.